# Characterisations of Europe's integrated water vapour and assessments of atmospheric reanalyses using more than two decades of ground-based GPS

Peng Yuan[1], Roeland Van Malderen[2], Xungang Yin[3], Hannes Vogelmann[4], Weiping Jiang[5], Joseph Awange[6], Bernhard Heck[1], Hansjörg Kutterer[1]

[1]Geodetic Institute, Karlsruhe Institute of Technology, Karlsruhe, 76131, Germany
[2]KMI-IRM, Royal Meteorological Institute of Belgium, Brussels, B-1180, Belgium
[3]NOAA National Centers for Environmental Information, Asheville, NC 28801, USA
[4]Karlsruhe Institute of Technology, IMK-IFU, Garmisch-Partenkirchen, 82467, Germany
[5]Wuhan University, GNSS Research Center, Wuhan, 430079, China
[6]Curtin University, School of Earth and Planetary Sciences, Perth, WA 6845, Australia

*Correspondence to*: Peng Yuan (peng.yuan@kit.edu)

**Abstract.** Ground-based Global Positioning System (GPS) has been extensively used to retrieve Integrated Water Vapour (IWV) and has been adopted as a unique tool for the assessments of atmospheric reanalyses. In this study, we investigated the multi-temporal-scale variabilities and trends of IWV over Europe by using IWV time series from 108 GPS stations for more than two decades (1994-2018). We then adopted the GPS IWV as a reference to assess six commonly used atmospheric reanalyses, namely CFSR, ERA5, ERA-Interim, JRA55, MERRA2, and NCEP2. The GPS results show that the peaks of the diurnal harmonics are within $15-21$ hour in local solar time at 90% of the stations. The diurnal amplitudes are $0-1.2$ kg m$^{-2}$ (0%-8% of the daily mean IWV), and they are found to be related to seasons and locations with different mechanisms, such as solar heating, land-sea breeze, and orographic circulation. However, mismatches in the diurnal cycle of ERA5 IWV between 09 and 10 UTC as well as between 21 and 22 UTC were found and evaluated for the first time and they can be attributed to the edge effect in each ERA5 assimilation cycle. The average ERA5 IWV shifts are -0.08 and 0.19 kg m$^{-2}$ at the two epochs and they were found to be more significant in summer and in the Alps, and in Eastern and Central Europe in some cases. Nevertheless, ERA5 outperforms the other reanalyses in reproducing diurnal IWV anomalies at all the 1-, 3-, and 6-hour temporal resolutions. ERA5 is also superior to the others in modelling the annual cycle and linear trend of IWV. For instance, the IWV trend differences between ERA5 and GPS are quite small, with a mean value and a standard deviation of 0.01% decade$^{-1}$ and 0.97% decade$^{-1}$, respectively. However, due to significant discrepancies with respect to GPS, CFSR and NCEP2 are not recommend for the analysis of IWV trends over Southern Europe and the whole Europe, respectively.

## 1 Introduction

Water vapour is the most important component of the Earth's atmosphere regarding the transport of energy by latent heat and radiative forcing. It is the most important gaseous source of infrared opacity in the atmosphere, and thus, the largest contributor to the natural greenhouse effect (Kiehl and Trenberth, 1997; Harries, 1997). It plays a key role in water and energy cycle (e.g, Trenberth and Fasullo, 2013), climate change (e.g, Schneider et al., 2010), and various weather and climate processes (e.g, Haar et al., 2012), and is crucial for the understanding of many extreme meteorological phenomena, such as atmospheric rivers (e.g, Zhu and Newell, 1994), hurricanes (e.g, Ejigu et al., 2021), floods (e.g, Turato et al., 2004),

and droughts and monsoons (e.g., Jiang et al., 2017; Fadnavis et al., 2021). However, due to its large spatiotemporal variability, its high-accuracy quantification remains a challenge.

Characterizing the multiple spatiotemporal scale variabilities and long-term trends of water vapour is of great importance for Europe, which is like a peninsula of the Eurasian landmass surrounded mainly by the Arctic Ocean, Atlantic Ocean, and Mediterranean Sea, to the north, west, and south, respectively. Owing to the moisture from the oceans carried by the prevailing westerlies, most of Western Europe has an oceanic climate with mild, wet, and turbulent weather in winter. In contrast, Southern Europe is characterised by a well-known dry-summer Mediterranean climate. Europe is generally vulnerable to the extreme events associated with abnormal water vapour transport and very sensitive to climate change (Field and Barros, 2014; Lavers et al., 2016), and has been the fastest warming up continent in recent decades (e.g., Copernicus, 2019). Since 1 K rise in temperature leads to a 7% increase of the water vapour holding capacity of the atmosphere as implied by the Clausius–Clapeyron equation (Trenberth et al. 2003), Europe's water vapour amount is noticeably increasing (Yuan et al., 2021). The increasing water vapour content fortifies the radiative forcing, leading to a higher temperature, which becomes most powerful if additional water vapour enters the upper troposphere and lower stratosphere (Solomon et al., 2010). The warmer and moister climate impacts all weather events and aggravates the risks of extreme events (Trenberth, 2012).

Atmospheric water vapour content can be expressed by using Integrated Water Vapour (IWV), which is defined as the total amount of water vapour present in a vertical atmospheric column from the Earth's surface to the top of the atmosphere in unit of kg m$^{-2}$ (Jones et al., 2020). The IWV is also known as the Total Column Water Vapour (TCWV) or Precipitable Water Vapour (PWV). At present, numerous techniques have been developed to measure water vapour, such as balloon-borne radiosondes (e.g., Durre et al., 2018; Kunz et al., 2013; Müller et al., 2016), aircraft measurements (e.g., Tilmes et al., 2010; Kunz et al., 2014; Krämer et al., 2020), satellite observations (e.g., Grossi et al., 2015; Beirle et al., 2018), and ground-based methods (e.g., Kämpfer 2012; Vogelmann et al., 2008). Global Navigation Satellite System (GNSS), represented by the USA's NAVSTAR Global Positioning System (GPS), has been exploited for water vapour retrieval by using its ground-based measurements (e.g., Bevis et al., 1992) or space-based radio occultation (e.g., Kursinski et al., 1995) since the 1990s. Although accurate vertical distribution of water vapour in the upper troposphere can be obtained by the GPS radio occultation (e.g., Randel and Wu, 2005; Randel et al., 2007), its accuracy is limited in the lower troposphere (e.g., Ao et al., 2003; Awange 2018) where the water vapour is most abundant. On the contrary, the ground-based GPS has proven to be an effective technique for IWV retrieval with advantages of high accuracy, high temporal resolution, and all-weather condition availability (Jones et al., 2020). The ground-based GPS has been utilized for IWV measurement in numerous global (e.g., Wang and Zhang, 2009; Vey et al., 2010; Chen and Liu, 2016) as well as regional (e.g., Bernet et al., 2020; Ejigu et al., 2021; Huang et al., 2021) studies. In addition, there have been nearly three decades of continuous GPS measurements with increasingly densified networks in Europe, such as the European Reference Frame (EUREF) Permanent Global Navigation Satellite System (GNSS) Network (EPN; Bruyninx et al., 2012). Given these benefits, ground-based GPS offers a unique tool to investigate the multiple spatiotemporal scale variabilities of IWV over Europe. The homogeneously reprocessed long-

term time series of GPS IWV is also quite beneficial to climate change studies (e.g., Van Malderen et al., 2020). GPS has increasingly been adopted to investigate various temporal features of Europe's IWV in many studies, such as diurnal cycle (e.g., Wang et al., 2009; Diedrich et al., 2016; Steinke et al., 2019), annual cycle (e.g., Parracho et al., 2018; Van Malderen et al., 2022), and trends (e.g., Nilsson and Elgered 2008, Ning et al., 2016, Wang et al., 2016). However, the multi-temporal-scale variabilities and trends of Europe's IWV have rarely been comprehensively studied using GPS.

Atmospheric reanalyses have been extensively adopted as the data source of IWV acquisition for the last several decades, owing to the benefits of regional/global coverage, consistent spatiotemporal resolution, and the availability of many other meteorological variables. However, their products may still be subject to large uncertainty. On the one hand, this is due to the fact that the reanalyses from different providers and different versions are inconsistent in the input data, assimilation schemes, besides using different physical schemes and representations. For example, the newly released fifth generation

global reanalysis from ECMWF (ERA5; Hersbach et al., 2020) has assimilated more datasets and instruments, which were not ingested in its predecessor ERA-Interim (ERAI; Dee et al., 2011). The assimilation system of ERA5 is also more advanced. On the other hand, systematic and random errors in the input data are unavoidable. For instance, the Integrated Global Radiosonde Archive (IGRA, Durre et al., 2018) provides a long record of relative humidity observations for the assimilation of reanalysis, but long-term radiosonde humidity measurements are very sensitive to changes in instrumentation

and measuring practice (McCarthy et al., 2009; Dai et al., 2011). Therefore, assessments of the reanalyses' water vapour products are indispensable for the accurate understanding and interpretations of Europe's weather and climate processes. The performances of various reanalyses' IWV products in Europe have been assessed by many regional/global studies using ground-based GPS data, as the GPS observations are not operationally assimilated by reanalyses (Hagemann et al., 2003; Bock et al., 2005; Heise et al., 2009; Vey et al., 2010; Alshawaf et al., 2018; Parracho et al., 2018; Wang et al., 2020; Yuan

et al., 2021). However, it is still quite hard to draw consistent conclusions on the performances of different reanalyses from these studies, as they are different in many aspects, such as the numbers and locations of the GPS stations, period of observations, data processing strategies, and performance metrics. A consistent assessment of various reanalyses by using homogeneously reprocessed long-term time series of GPS IWV as reference in Europe is still lacking. In addition, few studies have focused on the performance of the latest ERA5 in reproducing the temporal features of Europe's IWV so far.

In this paper, we focus on characterizing the multi-temporal-scale variabilities and trends of Europe's IWV by using more than two decades of GPS IWV, and then we assess the performances of six reanalyses' IWV products over Europe. The paper is organized as follows: In section 2, we describe the GPS and reanalyses datasets, IWV calculation methods, and consistency evaluation metrics. Section 3 assesses the consistency in daily time series and representativeness differences, while the diurnal variations and associated diurnal cycle, annual cycle, and long-term trends of Europe's IWV are

investigated using GPS in sections 4, 5, and 6, respectively. The performances of the reanalyses are also assessed accordingly, and the main findings are summarised in section 7.

## 2 Data and methods

### 2.1 GPS data

To characterise the IWV over Europe and assess the reanalyses, 1-hourly GPS IWV retrievals obtained from 108 stations are used (Fig. 1). Most of GPS stations are from the EPN network (Bruyninx et al., 2001). The GPS observations are from January 1994 to December 2018 with an average time length of 21 years and an average integration rate of 92% (Table S1). The integration rate is the ratio between the number of available daily IWV data points and the theoretical number of all possible observations in the time range. The IWV retrievals were estimated with GPS Zenith Total Delay (ZTD) provided by the Nevada Geodetic Laboratory (NGL; Blewitt et al. 2018). The GPS ZTD products from NGL were used because it covers a long time period of 25 years, so that it allows for better evaluations on the diurnal and annual climatological averages, and long-term trends. In comparison, another ZTD dataset from EPN-Repro2 (Pacione et al., 2017) only has about 19 years of data, ending in 2014.

The NGL processes the GPS data using the newly improved GipsyX v1.0 software in Precise Point Positioning (PPP) mode (Bertiger et al., 2020). Reprocessed orbits and clocks from Jet Propulsion Laboratory (JPL) Repro 3 were used together with the 2014 International GNSS Service (IGS) reference frame (IGS14; Rebischung and Schmid, 2016). The observations were weighted based on elevation ($e$) dependent function of $\sin e$. The cutoff elevation angle was set to 7°. The first order effect of the ionosphere was removed by employing ionosphere-free combinations of the GPS observations. The second order effect was corrected with International Geomagnetic Reference Field 12 (IGRF12; Thébault et al., 2015) and JPL's ionosphere maps. As for the modelling of tropospheric delay, the Vienna Mapping Function 1 (VMF1; Boehm et al., 2006) and its associated *a priori* Zenith Hydrostatic Delay (ZHD) were adopted.

### 2.2 Reanalysis data

The IWV derived from six commonly used global atmospheric reanalyses, namely, the newly released fifth generation global reanalysis (ERA5) and its predecessor ERA-interim (ERAI) from ECMWF, the Japanese 55-year Reanalysis (JRA55) from Japan Meteorological Agency (JMA), the Modern-Era Retrospective Analysis for Research and Applications, version 2 (MERRA2) from NASA Global Modeling and Assimilation Office (GMAO), and the Climate Forecast System Reanalysis (CFSR) and NCEP-DOE AMIP-II Reanalysis (NCEP2) from National Centers for Environmental Prediction (NCEP) are analyzed. The features of the reanalyses are summarised in Table 1. It is worth noting that JRA55 only provides humidity information at 27 pressure levels from 1000 to 100 hPa, though it has 37 levels in total. These six reanalyses are selected as their IWV products cover the period of the ground-based GPS data available since 1994. Despite ERAI having been decommissioned and superseded by ERA5 in August 2019, we still include it for the purpose of evaluating the progress of its successor ERA5. We therefore restricted the time range of all data records to a common period from 1994 to 2018.

**2.3 IWV retrievals**

The Zenith Total Delay (ZTD) estimates derived from GPS data processing can be converted into IWV, the total amount of water vapour present in a vertical atmospheric column from the Earth's surface to the top of the atmosphere in unit kg m$^{-2}$, using the following equation (Bevis et al., 1992):

$$\text{IWV} = \frac{10^6}{R_V \cdot [k_2' + k_3/T_m]} \cdot (\text{ZTD} - \text{ZHD}), \tag{1}$$

where $R_V$ denotes the gas constant for water vapour, $k_2'$ and $k_3$ are atmospheric refractivity constants (Bevis et al., 1994), and $T_m$ denotes the weighted mean temperature:

$$T_m = \frac{\int_{H_s}^{H_{top}} \frac{e}{T} dp}{\int_{H_s}^{top} \frac{e}{T^2} dp}, \tag{2}$$

where $e$ and $T$ are water vapour pressure and temperature profiles from the geopotential heights of the GPS station ($H_s$) to the top level of reanalysis ($H_{top}$), respectively.

The ZHD was modelled as follows (Saastamoinen, 1972; Davis et al., 1985):

$$\text{ZHD} = 2.2768 \frac{p_s}{1 - 2.66 \times 10^{-3} \cdot \cos(2\varphi_s) - 2.8 \times 10^{-7} H_s}, \tag{3}$$

where $p_s$ denotes the pressure at the GPS station with a latitude of $\varphi_s$ and a height of $H_s$. The $p_s$ and $T_m$ are computed by using the ERA5 pressure level product (see Yuan et al., 2021 and references therein). The ERA5 is selected as it can provide a 1-hourly product without temporal interpolation.

IWV values at the GPS stations are also calculated from the six reanalyses. In this calculation, the pressure level products of the reanalyses are used rather than their surface level IWV products, though it requires a heavier workload. This is because the GPS station and its nearby reanalysis surface grids are usually related to different heights. Vertical IWV adjustment is therefore usually required for the intercomparison between the IWV estimates from GPS and reanalyses. Compared to the reanalyses' surface level products, their pressure level products allow for a better characterisation of the vertical distribution of water vapour and tends to minimize the errors of the vertical adjustment (Parracho et al., 2018).

The scheme of the calculation is illustrated in Fig. 2. Assume a GPS station with a geopotential height of $H_s$ located between two adjacent reanalysis pressure levels ($p_j$ and $p_{j+1}$). We first determined its eight surrounding reanalysis grid nodes at these pressure levels ($N_i^{(j)}$ and $N_i^{(j+1)}$, $i$=1, 2, 3, and 4) and four auxiliary points ($A_i$). The auxiliary points are located at the GPS station height whereas their horizontal locations are identical to the associated reanalysis nodes. We then calculate the related meteorological variables at each auxiliary point. Exponential and linear interpolations are employed for the vertical corrections of pressure and temperature, respectively. If the auxiliary point is lower than the lowest pressure level (e.g., 1000 hPa), an extrapolation of these variables is conducted. For details of the vertical adjustment, readers are referred to Schüler (2001) and Wang et al. (2005). Next, we calculate the IWV at each auxiliary point by using a vertical integration:

$$\text{IWV}_i = \int_{H_s}^{H_{top}} \frac{q}{g} dp, \tag{4}$$

where $q$ and $g$ are specific humidity and gravitational acceleration profiles from $H_s$ to the reanalysis top level, respectively.

It is noteworthy that geopotential height system is employed in the reanalyses. Accordingly, the geopotential heights ($H_{gp}$) of the GPS stations rather than their ellipsoidal ($H_{el}$) or orthometric heights ($H_{or}$) are used in the above calculations. The conversion of height systems is carried out as follows (Dirksen et al., 2014; Wang et al., 2016; World Meteorological Organization, 2018):

$$H_{or} = H_{el} - N,$$

$$H_{gp} = \frac{\gamma_s(\varphi_s)}{9.80665} \cdot \frac{R(\varphi_s) \cdot H_{or}}{R(\varphi_s) + H_{or}}, \tag{5}$$

$$\gamma_s(\varphi_s) = 9.780325 \frac{1 + 1.93185 \times 10^{-3} \cdot \sin^2(\varphi_s)}{(1 - 6.69435 \times 10^{-3} \cdot \sin^2(\varphi_s))^{0.5}}, \tag{6}$$

$$R(\varphi_s) = \frac{6.378137 \times 10^6}{1.006803 - 6.706 \times 10^{-3} \cdot \sin(\varphi_s)^2}, \tag{7}$$

where the $N$ is the geoid heights in meter from the Earth Gravitational Model 2008 (EGM2008; Pavlis et al., 2012).

In the end, we estimate the IWV at the GPS station using a horizontal interpolation with an inverse distance weighting algorithm (Jade and Vijayan, 2008).

Representativeness differences arise in the comparison of the IWV derived by the ground-based GPS and reanalyses (Bock and Parracho, 2019). This is because local variations of IWV measured by the GPS might fail to be resolved by the reanalyses due to their coarse horizontal resolution. The representativeness differences can be evaluated statistically as proposed by Bock and Parracho (2019):

$$\delta_{\max} = max(|\text{IWV}_i - \text{IWV}_k|), \ i, k = 1,2,3,4, \ i \neq k, \tag{8}$$

where $\text{IWV}_i$ and $\text{IWV}_k$ are the IWV values of the horizontal auxiliary points $A_i$ and $A_k$, respectively (Fig. 2). Bock and Parracho (2019) found that the measure $\delta_{\max}$ was correlated with the standard deviation of the IWV differences between GPS and ERAI ($\sigma_\Delta$), and thus concluded that the representativeness differences between the IWV from GPS and ERAI contributed to their discrepancies. Here, we extend their work by estimating and comparing the representativeness statistics of the six reanalyses with various spatial resolutions (Table 1).

## 2.4 Pre-processing

The 1-, 3-, and 6-hourly IWV time series are screened by using a robust outlier detection method as follows (Yuan et al., 2021). For each data point, the data within a 30-day window centered at this point are extracted. The 25th percentile (Q1), 75th percentile (Q3), and interquartile range (IQR) of the subsequent time series are then calculated. Finally, the data point is identified as an outlier if it is outside the range of $[Q1 - 1.5 \times IQR, Q3 + 1.5 \times IQR]$. The IQR threshold and the 30-day sliding window are adopted as they allow for a good robustness and a proper accommodation for the natural variability of IWV. On average, about 1% data were filtered out by the data screening.

In this study, the performances of reanalyses IWV products in daily time series, diurnal cycle and variations, monthly annual cycle, and long-term linear trend are evaluated. As the temporal resolutions of the datasets are different, temporal interpolation and aggregation are conducted. To obtain daily mean IWV values, the 1-, 3-, and 6-hourly IWV time series are aggregated if they have at least 12, 4, and 2 data points in a day, respectively. The daily mean IWV time series at each station is further aggregated into monthly mean IWV series if there are at least 15 data points available in a month.

## 2.5 Homogenisation

The GPS IWV time series can be inhomogeneous due to changes in GPS data processing strategies and station-related changes like hardware changes or changes in the electromagnetic environment (Van Malderen et al., 2020; Nguyen et al., 2021 and references therein). We employed a homogenisation approach as described in Appendix A with step-by-step details and examples at two stations (HERS and ERLA).

Here, we only provide a brief introduction on the approach. We first avoided inhomogeneities due to changes in GPS data processing strategy by using the homogeneously reprocessed GPS ZTD product. We then homogenised the GPS IWV time series by using the RHtestsV4 software (Wang and Feng, 2013). This software is developed especially for the detection and adjustment of changepoints in climatic time series, and it has been used in the homogenisation of IWV time series in previous studies (Ning et al., 2016; Schröder et al., 2016; Van Malderen et al., 2020).

We took the IWV time series from all the six reanalyses as references for the GPS IWV homogenisation and used a strategy to avoid the impacts of possible changepoints in individual reanalyses. However, we did not homogenise the reanalyses IWV time series because they represent the native quality of the reanalyses that we would like to assess. In addition to matching the detected changepoints with inspecting GPS metadata information (GPS station log files and IGSMAIL), we also allowed for possible undocumented changepoints in the GPS IWV time series.

## 2.6 Metrics for consistency evaluation

To evaluate the performances of the reanalyses in reproducing IWV, Kling-Gupta Efficiency (KGE) is employed as a metric. The KGE is a composite index introduced by Gupta et al. (2009) and modified by Kling et al. (2012). It takes bias, variability, and correlation into account in the equation below:

$$\text{KGE} = 1 - \sqrt{(\beta - 1)^2 + (\gamma - 1)^2 + (r - 1)^2}, \tag{9}$$

with

$$\beta = \frac{\mu_R}{\mu_G}, \tag{10}$$

$$\gamma = \frac{CV_R}{CV_G} = \frac{\sigma_R/\mu_R}{\sigma_G/\mu_G}, \tag{11}$$

and $\mu_R$, $\sigma_R$, and $CV_R$ are the mean value, standard deviation, and coefficient of variation of the reanalysis IWV time series, respectively. The $\mu_G$, $\sigma_G$, and $CV_G$ are the corresponding parameters for GPS. The $\beta$ and $\gamma$ indicate the consistencies in the

mean and variability, respectively. In case of $\mu_R$ and $\mu_G$ being equal to zero, such as for the diurnal anomalies in Section 4.3, $\gamma = \frac{\sigma_R}{\sigma_G}$. The $r$ indicates the Pearson correlation coefficient between the GPS and reanalyses time series. With perfect consistencies in mean, variability, and correlation, the values of $\beta$, $\gamma$, and $r$ are identical to 1, respectively. From Eq. (9), it follows that a larger KGE score indicates a better consistency. Ideally, the KGE metric reaches its maximum of 1.

In addition, to be comparable to some previous studies, we also used the Root Mean Square estimation of the IWV differences between the reanalyses and GPS:

$$\text{RMS}_\Delta = \sqrt{\frac{\sum_{i=1}^{n}(\text{IWV}_R - \text{IWV}_G)^2}{n}},\tag{12}$$

where $\text{IWV}_R$ and $\text{IWV}_G$ are the IWV time series at a specific station from reanalysis and GPS, respectively, and $n$ is number of data points in the IWV time series.

## 3 Assessments of daily time series

### 3.1 Representativeness differences

Figure 3 compares the standard deviations of daily IWV difference ($\sigma_\Delta$) and the representativeness statistic ($\delta_{\max}$) for the six reanalyses, respectively. The first point to note is that the $\sigma_\Delta$ and $\delta_{\max}$ are strongly correlated for all the reanalyses, with $r$ values from 0.76 for NCEP2 to 0.90 for ERA5. The result is in line with Bock and Parracho (2019), who compared
ERAI to a global GPS network. Figure 3 indicates that the representativeness differences contribute to the discrepancies between the reanalyses and GPS. The larger $\sigma_\Delta$ and $\delta_{\max}$ values are found at stations close to mountain or sea, such as BZRG (Bolzano, Italy, 11.34 °E, 46.50 °N) and TORI (Torino, Italy, 7.66 °E, 45.06 °N) at the foothills of the Alps, and ALME (Almería, Spain, 2.46 °W, 36.85 °N) near the west coast of Mediterranean Sea. On the whole, ERA5 is characterised by the lowest $\sigma_\Delta$ and $\delta_{\max}$, with values of 0.5−1.6 and 0.2−2.1 kg m$^{-2}$, respectively. In contrast, NCEP2 has the largest $\sigma_\Delta$
and $\delta_{\max}$, with values of 1.1−3.0 and 1.8−5.2 kg m$^{-2}$, respectively. The difference could be due to the fact that the spatiotemporal resolution of ERA5 is much higher than NCEP2 as indicated in Table 1. This result indicates that ERA5, with improved spatiotemporal resolution and data assimilation, is capable of reducing the discrepancy and representativeness difference with respect to GPS.

### 3.2 Assessments using KGE

Figure 4 shows the geographical distributions of the ratio of mean $\beta$, the ratio of coefficient of variation $\gamma$, the correlation $r$, and the synthetic KGE metric for the six reanalyses by taking the daily GPS IWV time series as references. Moreover, the statistics of these scores are displayed in Fig. 5 with box-and-whisker plots. From the first column of Fig. 4 and Fig 5a we can see that $\beta$ scores are slightly larger than 1 for all the reanalyses except JRA55, indicating a general wet bias with respect to the GPS IWV. For example, MERRA2 has the largest median $\beta$ with a value of 1.04, indicating a

general wet bias of 4%. Only JRA55 scores a median $\beta$ smaller than 1 (0.99, Fig. 5a), indicating a slight dry bias of 1%. The wet and dry biases in the reanalyses have been partly reported by several previous studies. For instance, Schröder et al. (2018) concluded that CFSR, ERAI, and MERRA2 are too moist over Europe compared to the ensemble mean of various satellite and reanalysis IWV records, whereas JRA55 has negligible bias there. The wet bias in ERAI over Europe was also noted by Parracho et al. (2018). Analysing the reasons of the biases is of potential interest; however, it would go beyond the scope of this paper.

Regarding the consistency in variability, most of the $\gamma$ scores are less than 1, with the associated median values ranging between 0.96 and 0.98 (Fig. 5b). The results indicate a good reproduction of daily IWV variability by the reanalyses, albeit with a slight underestimation. The correlations are also pretty good with median values larger than 0.97 (Fig. 5c). However, the scores in variability and correlation are lower for the coastal stations, as shown in the second and third columns of Fig. 4, respectively. In particular, the $\gamma$ and $r$ of NCEP2 are less than 0.96 for some coastal stations located at Western and Southern Europe. This could be explained by the representativeness differences that impact the consistency of the IWV from reanalyses and GPS in their variabilities. For a coastal GPS station with an on-site measurement of IWV, a part of its associated four reanalysis grid nodes may be located over the sea whereas others over land. As a consequence, the representativeness difference for such a coastal site is more severe than inland stations surrounded with flat terrain due to land-sea thermal contrast (Drobinski et al., 2018).

Figure 5d compares the overall consistency in daily IWV evaluated using KGE. It can be seen that ERA5 is characterised by the largest median KGE (0.97). The result reveals the superiority of ERA5 over the other reanalyses. Moreover, comparing $\beta$, $\gamma$, and $r$ (Fig. 5a−c) shows that $r$ tends to contribute the least to the overall inconsistency, indicating that improving the consistencies in mean values and variabilities are more important for the daily IWV time series.

## 4 Assessments of diurnal variations

Despite the atmospheric water vapour being quite unstable, it is characterised by a diurnal cycle. The diurnal IWV cycle can be driven by different mechanisms, such as evapotranspiration and condensation related to temperature, solar heating, and underlying surface conditions, as well as advection of air at different spatial scales (Dai et al., 2002; Diedrich et al., 2016; Koji et al., 2022).

In this section, the diurnal variations of IWV are investigated in two aspects: diurnal cycle and diurnal anomaly. The diurnal anomalies ($IWV_{DA}$) are calculated as follows (Diedrich et al., 2016; Steinke et al., 2019):

$$IWV_{DA} = IWV - \overline{IWV}, \tag{13}$$

where IWV is the 1-hourly IWV time series in one day and $\overline{IWV}$ is the associated daily mean.

Each station's diurnal cycle is computed by averaging its diurnal anomalies for each season separately and for the entire time span. Diurnal and semidiurnal harmonics are calculated with Least Squares Estimation (LSE) based on the seasonal and all-time averaged diurnal anomalies:

$$\text{IWV}(t) = \sum_{i=1}^{2} A_i \sin\left(\frac{2\pi}{24} \cdot i \cdot t + \varphi_i\right) \tag{14}$$

where $t$ is Local Solar Time (LST) in hours, $A_i$ and $\varphi_i$ are the amplitudes and phases of the diurnal ($D_1$) and semidiurnal

($D_2$) cycles, respectively. The phases are then adjusted to the at the peaks of associated harmonics, with $\varphi_1 \in [0,24)$ and $\varphi_2 \in [0,12)$ LST in hours.

## 4.1 Diurnal GPS IWV cycle

Starting with the all-time averaged amplitudes of the diurnal GPS IWV harmonic shown in Fig. 6e, two remarkable values can be first noted at stations NICO (Nicosia, Cyprus, 33.14 °N, 33.40 °E, 161.9 m) and ZECK (Zelenchuksky, Russia,

43.79 °N, 41.57 °E, 1143.4m) with values of 1.2 and 1.0 kg m⁻², respectively. Moreover, the diurnal harmonics at the Mediterranean Coast are generally stronger than the other regions in Europe (0.5−0.8 versus 0−0.5 kg m⁻²). Obvious seasonal differences can also be seen in their diurnal harmonics, with significantly larger amplitudes in summer (June, July, and August; JJA) than the other seasons due to the stronger solar heating effect with minimal cloud coverage in Mediterranean summer (Enriquez-Alonso et al., 2016). However, the semidiurnal harmonics are much weaker, and their seasonal variations

are less significant (Fig. 6f−j). The all-time averaged semidiurnal amplitudes are lower than 0.22 kg m⁻², except the two stations NICO and ZECK with values of 0.4 and 0.3 kg m⁻², respectively. The ratios between the all-time averaged semidiurnal and diurnal amplitudes are lower than 30% at 88% of the stations (95/108). As for the phases, the diurnal and semidiurnal terms are generally consistent over seasons (Fig. k−t), and their all-time averaged peaks are within 15−21 LST and 01−06 LST at 90% of the stations (97/108), respectively.

In order to compare the characteristics of diurnal IWV amplitudes at different stations, we calculated each station's relative amplitude as the ratio between their respective (semi-) diurnal amplitudes and mean IWV as displayed in Fig. 7a and 7d. We classified the GPS stations into three types according to their geographical characteristics (Fig. 7a) and analysed their relationships with each station's altitude and distance-to-sea (SeaDist). Firstly, we divided the stations with a limit of 20 km on their SeaDist. We further separated the stations located at Mediterranean Coast (MedCoast; SeaDist<20 km, 32 °N

<Lat<46 °N, 5 °W <Lon<45 °E) from the other coastal (OtherCoast) stations, because their characteristics are quite different as can be seen from Fig. 7a. Consequently, the 108 stations are classified into 62 Inland stations, 12 MedCoast stations, and 34 OtherCoast stations.

As can be seen from Fig. 7a−c, all the OtherCoast stations are lower than 300 m and their relative diurnal IWV amplitudes are the weakest, with a range from 0.3% to 1.9% and a median of 1.1%. Within the altitude limit of 300 m, the

Inland stations are characterised with moderately larger diurnal amplitudes (1.5%−2.5%). The results indicate the effect of land-sea breeze circulation on mitigating the intensity of diurnal IWV cycle at the Atlantic coasts of Europe with respect to

the Inland Europe. However, the land-sea breeze effect can be less significant for the MedCoast stations, because their diurnal amplitudes are significantly larger (1.1%−4.2%) than the OtherCoast stations. The stronger diurnal IWV cycles at the MedCoast stations can be explained by the stronger solar heating effect at the Mediterranean Coast than the other European coasts, especially during summer daytime under stable and clear sky weather condition. In addition, it can be seen from Fig. 7b and 7e that the relative diurnal and semidiurnal IWV amplitudes are well correlated with altitudes, with correlation coeffeicents of 0.66 and 0.67, respectively. This relationship indicates the effect of orographic circulation, which can enhance the diurnal range of temperature at higher altitudes (Diedrich et al., 2016).

In addition, we selected six stations with various altitudes, SeaDist, and climates to illustrate the diversity of diurnal IWV cycles in Europe as shown in Fig. 8. The climate zones of the GPS stations are classified according to Köppen Climate Classification (Beck et al., 2018) and the properties of the stations are listed in Table S2.

Station NICO is located on Cyprus Island with hot semi-arid climate (BSh). Despite only 21.5 km far away from coastline, NICO has the largest diurnal IWV amplitude with a value of 1.2 kg m$^{-2}$, equivalent to a relative amplitude of 6.7%. This is mainly due to the large diurnal temperature range in Cyprus, especially in summer (Price et al., 1999).

Both the MedCoast station VALE (Valencia, Spain, 39.48 °N, 0.34 °W, 27.0 m) and the OtherCoast station NEWL (Newlyn, UK, 50.10 °N, 5.54 °W, 11.0 m) are very close to coastline, with SeaDist values of 1.2 and 0.5 km, respectively. However, their diurnal amplitudes are quite different (absolutely 0.8 versus 0.1 kg m$^{-2}$, relatively 3.9% versus 0.7%). As explained earlier for the difference between the two station types, their weather conditions are different. VALE is located at Mediterranean Coast with cold semi-arid (BSk) climate. Its strong diurnal cycle, especially in summer with an amplitude of 1.4 kg m$^{-2}$ (4.6%), is attributed to intense solar heating under minimal cloud coverage weather condition. In contrast, NEWL is located at the coast of the English Channel with temperate oceanic (Cfb) climate, and its smaller diurnal amplitude can be due to the weaker solar heating effect under the unstable and cloudy weather condition, in addition to the land-sea breeze effect on mitigating diurnal temperature range.

Station ZECK is in the Greater Caucasus with humid continental (Dfb) climate. Its diurnal amplitude is much stronger than PENC (Penc, Hungary, 47.79 °N, 19.28 °E, 248.3 m) with the same climate (1.0 kg m$^{-2}$ and 7.9% versus 0.2 kg m$^{-2}$ and 1.5%). As ZECK is much higher than PENC, their difference in amplitude is consistent with the pattern for most Inland stations, which can be explained by the effect of orographic circulation (Diedrich et al., 2016). In addition, KIR0 (Kiruna, Sweden, 67.88 °N, 21.06 °E, 469.3 m) is typical for many stations in North Europe. Although its diurnal cycle is quite weak with an amplitude of only 0.1 kg m$^{-2}$ (1.7%), it is well fitted by the sinusoidal harmonic curve fitting.

## 4.2 Mismatches in diurnal ERA5 IWV cycle

Only the diurnal IWV cycle from the 1-hourly ERA5 time series was evaluated with respect to GPS, as the temporal resolutions of the other reanalyses are too coarse to characterise the diurnal cycle. However, we found significant shifts in the diurnal IWV anomalies between 09 and 10 (10-09) UTC as well as between 21 and 22 (22-21) UTC at part of the stations, such as NEWL and PENC displayed in Fig. 8. The ERA5 developers have noticed such mismatches in the diurnal

cycles of individual meteorological variables, such as its near surface wind, temperature, and humidity products (see Known Issues 8 and 9 in https://confluence.ecmwf.int/display/CKB/ERA5%3A+data+documentation#ERA5:datadocumentation-Knownissues), and the problem is attributed to the edge effect in each ERA5 assimilation cycle (from 10 to 21 UTC and from 22 to 09 UTC +1 day). However, according to our knowledge, the magnitude of the mismatch in the diurnal cycle of ERA5 IWV and its spatiotemporal characterisations have not been investigated yet. Therefore, we will quantify and analyse the mismatch in the ERA5 IWV over Europe in this section.

Figure 9 compares the diurnal IWV cycle from ERA5 and GPS for each season at station PENC. From this figure, we can derive that there are no mismatches in the diurnal GPS IWV cycle, although the GPS IWV (slightly) relies on $T_m$ (hence humidity and temperature) from ERA5 for the ZTD to IWV conversion, as shown in Eq. (2). GPS IWV estimates are therefore regarded as reference data to evaluate the mismatches in the diurnal cycle of ERA5 IWV. At station PENC, the mismatches in ERA5 are seasonal dependent, which are strongest in summer (JJA) but weakest in winter (DJF).

Figure 10 compares the shifts at 10-09 UTC and 22-21 UTC from ERA5 and GPS at all the stations, respectively. The shifts in the GPS IWV cycle are regarded as reference, representing the natural IWV changes at the two epochs. As can be seen from Fig. 10a−e and 10k−o, the ERA5 IWV series generally drop from 09 to 10 UTC and then jump from 21 to 22 UTC. The ERA5 artificial shifts at the two epochs are most significant in summer, with average values of -0.23 and 0.35 kg m$^{-2}$, respectively. In contrast, the average natural shifts in summer estimated from GPS IWV are only 0.11 and -0.08 kg m$^{-2}$ respectively. Moreover, the all-time averaged natural shifts in GPS IWV are only 0.05 and -0.05 kg m$^{-2}$, respectively. However, the artificial shifts in ERA5 IWV are -0.08 and 0.19, respectively. As can be seen from the geographic distributions of the shifts shown in Fig. 10e and 10o, the ERA5 shifts at 10-09 UTC are most significant at the Alps and Eastern Europe, whereas the shifts at 22-21 UTC are more widespread in Central Europe. The reasons for their geographical patterns are unknown and needs further investigation. Since the average diurnal amplitude of the reference GPS IWV is only 0.32 kg m$^{-2}$, the artificial shifts in ERA5 IWV cannot be ignored when analysing the diurnal IWV cycle in these regions.

## 4.3 Diurnal anomalies

Diurnal anomaly represents high-frequency variations of IWV, associated with weather phenomena like heavy rainfall. Therefore, evaluations on the consistencies between the IWV diurnal anomalies from reanalyses with respect to GPS are conducive to a better understanding on their performances in extreme weather, especially for ERA5 with a significantly enhanced temporal resolution of 1-hour.

We first evaluated all the reanalyses with respect to GPS at 1-hour temporal resolution. For the reanalyses with coarser resolutions (3-hour and 6-hour), we interpolated their time series to 1-hour by using cubic spline, which is slightly superior to linear interpolation. Statistics of the evaluation results are listed in Table 2. At the temporal resolution of 1-hour, ERA5 scores the highest average $\gamma, r,$ and KGE with values of 0.98, 0.89, and 0.88, respectively. The results indicate that the 1-hourly ERA5 diurnal anomalies has the best agreement with GPS in variability and correlation. However, NCEP2 performs the worst, as it scores the lowest $\gamma, r,$ and KGE with values of 0.75, 0.69, and 0.60, respectively. To be comparable to many

other studies, we also evaluated the consistencies by using the more commonly used indicator — RMS of IWV difference ($RMS_\Delta$) as shown in Eq. (12). Results show that ERA5 and NCEP2 score the lowest and largest average $RMS_\Delta$ values of 0.97 and 1.57 kg m$^{-2}$, respectively. Therefore, we drew the same conclusion as from the KGE scores. The evaluations confirm the superiority of the 1-hourly ERA5 over the other reanalyses, most likely because of its enhanced spatiotemporal resolutions and other improvements in data assimilation.

For a fairer intercomparison, we also evaluated ERA5 and the other reanalyses at their respective native resolutions. Accordingly, we extracted the ERA5 IWV every 3 and 6 hours. The comparison with MERRA2 at its native resolution of 3-hour shows that, ERA5 achieves larger KGE (0.89 versus 0.86) and smaller $RMS_\Delta$ (0.97 versus 1.15 kg m$^{-2}$). Moreover, the average KGE value of the 6-hourly ERA5 IWV is 0.89, which is higher than CFSR, ERAI, JRA55, and NCEP2, with values of 0.86, 0.83, 0.79, and 0.60, respectively. The results indicate that ERA5 is still superior to the other reanalyses at the coarser temporal resolutions.

## 5 Assessments of annual cycle

We analysed the annual cycles of the homogenised monthly GPS IWV time series as shown in Fig. 11a with increasing latitude from bottom up. Most of the annual IWV cycles reach their maxima in July and August, with peak values from 17.1 to 32.2 kg m$^{-2}$. In contrast, the minima of the annual cycles are typically in January and February, with values from 4.2 to 17.1 kg m$^{-2}$. The maximum and minimum values of the annual cycles are generally increasing with decreasing latitude (Fig. 10a) and decreasing altitude (not shown).

Figure 11 b−g present the differences between the annual cycle of IWV estimated by the reanalyses with respect to GPS. It can be seen that ERA5 has the least differences. Indeed, the quantitative evaluation shows that ERA5 obtains the highest median KGE score having a value of 0.97 (Fig. 5h), indicating that it outperforms the other reanalyses. CFSR ranks last (median KGE=0.94) due to its overestimation of mean values (median $\beta$=1.04) and underestimation of variability (median $\gamma$=0.96). Moreover, although the consistencies between the annual IWV signals from reanalyses and GPS are generally rather good, there could be significant discrepancies in specific seasons and regions. For example, JRA55 has an average dry bias of 0.5 kg m$^{-2}$ with respect to the GPS IWV from May to September at the stations in the south (32 °N−48 °N, Fig. 11e), whereas NCEP has obvious wet biases (0.7−2.4 kg m$^{-2}$) in the annual average of IWV with respect to the GPS results at low latitudes (32 °N−40 °N, Fig. 11g). In addition, significant biases (1.2−5.9 kg m$^{-2}$) from March to September can be seen at station BZRG in the evaluations of CFSR, ERAI, and MERRA2. The discrepancies are most likely related to the remarkable representativeness differences between the reanalyses and GPS at BZRG (Fig. 3), which is located in the Alps.

## 6 Assessments of linear trends

We carried out a homogenisation of the GPS IWV time series by using the RHtestsV4 software (Wang and Feng, 2013) before the analysis. The software is dedicated to the homogenisation of climatic time series. In addition, we adopted a homogenisation strategy which allows for changepoints with and without support from metadata. We took all the six reanalyses as references and attempted to avoid the impacts of possible changepoints in specific reanalyses. However, we did not homogenise the reanalyses IWV time series because they represent the native quality of the reanalyses that we would like to assess. The homogenisation approach is detailed in Appendix A.

We then estimated the linear IWV trends from all the six reanalyses and the homogenised GPS IWV time series after the removal of annual cycle. In order to obtain realistic uncertainties of the trend estimates, we analysed the time series by using the Hector software version 1.7.2 (Bos et al.,2019). We tested four commonly used noise models, namely White Noise (WN), first-order AutoRegressive AR(1), AutoRegressive Moving Average ARMA(1,1), and Power-Law noise (PL). We then selected the optimal model of each time series by using Bayesian information criterion (BIC; Schwarz, 1978). Readers are referred to Yuan et al. (2021) for more details. The IWV trend estimates, associated uncertainties, and specific optimal noise models are listed in Table S3.

Figure 12a shows the GPS IWV trends after the homogenisation. The IWV trends generally increase from the Atlantic coasts towards southeast. The average trends of the OtherCoast, Inland, and MedCoast stations are 0.2, 0.5, and 0.9 kg m$^{-2}$ decade$^{-1}$, respectively. The corresponding relative IWV trends are 1.7%, 3.1%, and 5.0% per decade (Fig. 12d), respectively. The largest trend is found at station TUBI (Gebze, Turkey, 29.5 °E, 40.8 °N), with a value of 1.6 kg m$^{-2}$ decade$^{-1}$ (8.7% decade$^{-1}$). As can be seen from Fig. 12b, AR1 is the optimal noise model at 58% of the stations (63 out of 108), which are mostly located close to sea or in Northern Europe. Most parts of Central Europe are characterised by WN, whereas Belgium and central Germany are best modelled by ARMA11. In addition, PL is superior to the other models at four stations. The optimal noise models for the monthly IWV time series of the six reanalyses show similar geographical patterns as the GPS IWV. The geographical patterns of the optimal noise models obtained in this study are different from Yuan et al. (2021), although the geographical patterns of the IWV trends are consistent to previous studies (Parracho et al., 2018; Nguyen et al., 2021; Yuan et al., 2021). It is most likely due to the difference that monthly IWV time series are used here, whereas daily IWV series were used in that work.

We evaluated the IWV trends derived from the six reanalyses by taking the trends from the homogenised GPS IWV time series as reference. We compared the difference in relative IWV trend rather than the absolute trend because the relative trend is not affected by the IWV bias in individual reanalysis. As can be seen in Fig. 13, ERA5 shows the best agreement in IWV trends with respect to GPS, with a mean value and a standard deviation of 0.01% decade$^{-1}$ and 0.97% decade$^{-1}$ in their trend differences, respectively, indicating an improvement compared to its predecessor ERAI (-0.04±1.15% decade$^{-1}$). JRA55 also only have an average trend difference as low as 0.01% decade$^{-1}$ but with a slightly larger standard deviation (1.12% decade$^{-1}$) than ERA5. Compared to the GPS IWV trends, MERRA2 has an underestimation of 0.22% decade$^{-1}$ on

average (Fig. 13e), whereas CFSR resulted in an overestimation by 0.22% decade$^{-1}$ (Fig. 13a). The standard deviation of the CFSR-GPS IWV trend differences is as large as 1.73 % decade$^{-1}$, which is mainly caused by the significant differences from -5.1% decade$^{-1}$ to 4.9% decade$^{-1}$ in South Europe (Fig. 13a). The results suggest that the CFSR IWV trends is less accurate in Southern Europe and should be carefully validated before climate change analysis.

In addition, Fig. 13f shows that the mean value and standard deviation of NCEP2-GPS IWV trend differences are 1.61% decade$^{-1}$ and 1.86% decade$^{-1}$, respectively, indicating a general overestimation of IWV trends from NCEP2 with respect to GPS. In particular, the average NCEP2-GPS IWV trend differences is as large as 2.9% decade$^{-1}$ at 31 stations located in Southern Europe (32 °N <Lat<46 °N, 10 °W <Lon<25 °E). However, NCEP2 underestimates the trends by at two stations in Eastern Mediterranean, with an average difference of -3.1% decade$^{-1}$ compared to GPS IWV. Therefore, we concluded that the NCEP2 IWV trends are not qualified for climate change analysis in Europe.

## 7 Conclusions

In this study, Integrated Water Vapour (IWV) time series for the period 1994−2018 were retrieved from continuous GPS observations at 108 ground-based GPS stations in Europe, with an average period of 21 years for those time series. The temporal features of Europe's IWV, such as its diurnal cycle and variation, annual cycle, and linear trend were then investigated. Moreover, the performances of six frequently used global atmospheric reanalyses in Europe were assessed for the first time, namely CFSR, ERA5, ERA-Interim (ERAI), JRA55, MERRA2, and NCEP2. The main findings are summarised here below.

(i) The agreement between the daily GPS IWV time series and the six reanalyses are found to be best for ERA5 and worst for NCEP2, with standard deviations of IWV differences of 0.5−1.6 and 1.1−3.0 kg m$^{-2}$, respectively. The standard deviations of IWV differences are well correlated with representativeness statistics of the six reanalyses, indicating that the representativeness differences contribute to the discrepancies between the reanalyses and GPS.

(ii) The diurnal amplitudes are 0−1.2 kg m$^{-2}$, account for 0%-8% of the associated daily mean IWV. The semidiurnal amplitudes are weaker, with values lower than 30% of the diurnal amplitudes at 88% of the stations. The peaks of the diurnal and semidiurnal harmonics are within 15−21 LST and 01−06 LST at 90% of the stations, respectively. The diurnal amplitudes are larger at the Mediterranean Coast (1.1%−4.2%), which is most likely because of stronger solar heating effect. In comparison, the diurnal amplitudes at other coastal regions in Europe are lower (0.3%−1.9%), which can be due to land-sea breeze and weaker solar heating effect. In addition, the relative diurnal and semidiurnal IWV amplitudes are correlated with altitudes with correlation coefficients of 0.66 and 0.67, respectively, which can be related to orographic circulation.

(iii) Mismatches in the diurnal cycle of ERA5 IWV product were found and evaluated between 09 and 10 UTC as well as between 21 and 22 UTC. The problem can be attributed to the edge effect in each ERA5 assimilation cycle, and it has been noticed in some other meteorological variables provided by ERA5. The average artificial shifts in ERA5 IWV are -0.08 and 0.19 kg m$^{-2}$ at the two epochs, respectively. In contrast, the natural shifts in GPS IWV are 0.05 and -0.05 kg m$^{-2}$,

respectively. The ERA5 shifts are dependent on seasons and locations. The ERA5 shifts are more significant in summer than in winter. Moreover, the ERA5 shifts from 09 to 10 UTC are most significant at the Alps and Eastern Europe, whereas the shifts from 21 to 22 UTC are more widespread in Central Europe. As the average diurnal IWV amplitude obtained from GPS is only 0.32 kg m$^{-2}$, the artificial shifts in ERA5 IWV cannot be ignored in diurnal IWV cycle analysis in these regions.

(iv) Regarding diurnal IWV anomalies, ERA5 shows the best consistency with GPS at a temporal resolution of 1-hour, with an average RMS of IWV difference (RMS$_\Delta$) of 0.97 kg m$^{-2}$, whereas the average RMS$_\Delta$ are 1.14−1.57 kg m$^{-2}$ for the others five reanalyses. ERA5 also outperforms the other reanalyses at their respective resolutions (3-hour and 6-hour), indicating the benefits from its enhanced spatial resolutions and other improvements in data assimilation, in addition to its higher temporal resolution.

(v) All the monthly IWV time series are modulated with apparent annual cycles, with minima in January and February (4−17 kg m$^{-2}$) and maxima in July and August (17−32 kg m$^{-2}$). The maxima and minima of the annual cycles show consistent geographical patterns, with larger values towards the equator and at lower altitude sites. ERA5 ranks the first in modelling the annual cycles of IWV (median KGE=0.97) with respect to GPS IWV, CFSR ranks last (median KGE=0.94) due to its overestimation of mean values (median $\beta$=1.04) and underestimation of variability (median $\gamma$=0.96).

(vi) Europe's IWV is increasing as observed from more than two decades of continuous GPS observations. The trends generally increase from the Atlantic Coast towards southeast. The average trends of the Atlantic coasts, Inland Europe, and Mediterranean coasts are 0.2, 0.5, and 0.9 kg m$^{-2}$ decade$^{-1}$, which are equivalent to relative values of 1.7%, 3.1%, and 5.0% per decade, respectively. The monthly IWV time series are best modelled by first-order AutoRegressive (AR1) noise at most stations located close to sea and Northern European, whereas AutoRegressive Moving Average ARMA(1,1) and White Noise (WN) and ARMA11 models are preferred at the rest stations. Power-Law noise (PL) is found to be optimal at only four stations. As for the performances of the reanalyses in reproducing IWV trends, ERA5 achieves the best consistency with GPS IWV trends, with a mean value and a standard deviation of 0.01% decade$^{-1}$ and 0.97% decade$^{-1}$ in their trend differences, respectively. However, CFSR is not qualified for the analysis of IWV trends in Southern Europe, due to its significant discrepancies with respect to GPS. The NCEP2 IWV trends over the whole Europe are also not recommend for the same reason.

Overall, it can be concluded that the reanalyses successfully reproduce the spatiotemporal IWV variability over Europe, as assessed by using the GPS IWV dataset, with ERA5 slightly outperforming the other reanalyses at most temporal scales. It is noteworthy that the pressure and weighted mean temperature obtained from ERA5 were used in the conversion of GPS ZTD to IWV due to lack of long-term accurate and complete *in-situ* meteorological observations at the GPS stations. This could partly contribute to the superior agreement between the IWV from ERA5 and GPS. Future studies could validate the IWV products of reanalyses using ground-based GPS when independent, quality-assured, and complete meteorological observations at the GPS stations are available. IWV measurements from other techniques could also be helpful.

*Data availability.* The GPS ZTD data were provided by NGL (http://geodesy.unr.edu). The ERA5 and ERAI data were
downloaded from the climate data store (https://cds.climate.copernicus.eu) and ECMWF (https://www.ecmwf.int),
respectively. The CFSR and JRA55 data are available in the research data archive of NCAR (https://rda.ucar.edu). MERRA2
data were derived from NASA Goddard earth sciences data and information services center (https://disc.gsfc.nasa.gov).
NCEP2 data were obtained from NOAA/OAR/ESRL physical sciences laboratory (https://psl.noaa.gov). The GNSS IWV
time series are available from the corresponding author upon reasonable request.

*Author contributions.* PY: Conceptualisation, Methodology, Formal analysis, Investigation, Writing Original Draft, and
Visualisation. RV, XY, HV, WJ, JA, and BH: Investigation and Reviewing. HK: Investigation, Reviewing, Supervision,
Project administration, and Funding acquisition.

*Competing interests.* The authors declare that they have no competing interests.

*Acknowledgements.* We are grateful to NGL for providing the GPS ZTD products and many institutions for sharing the
continuous GPS observations. We thank ECMWF, JMA, NASA GAMO, and NCEP for providing the reanalyses products.
This work was performed under the German Research Foundation (DFG, project number: 321886779) and the Program for
Hubei Provincial Science and Technology Innovation Talents of China (grant number: 2022EJD010).

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

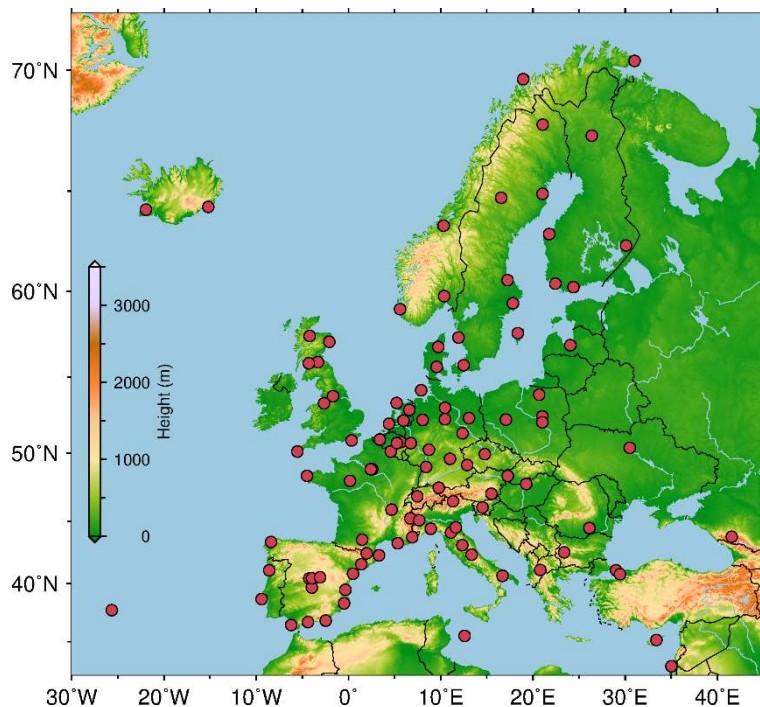

**Figure 1.** Geographical distribution of the 108 GPS stations (red dots). An enlarged version of this figure with station names is provided in Fig. S1 in the supplementary material. The coordinates of the stations and their time lengths are provided in
Table S1.

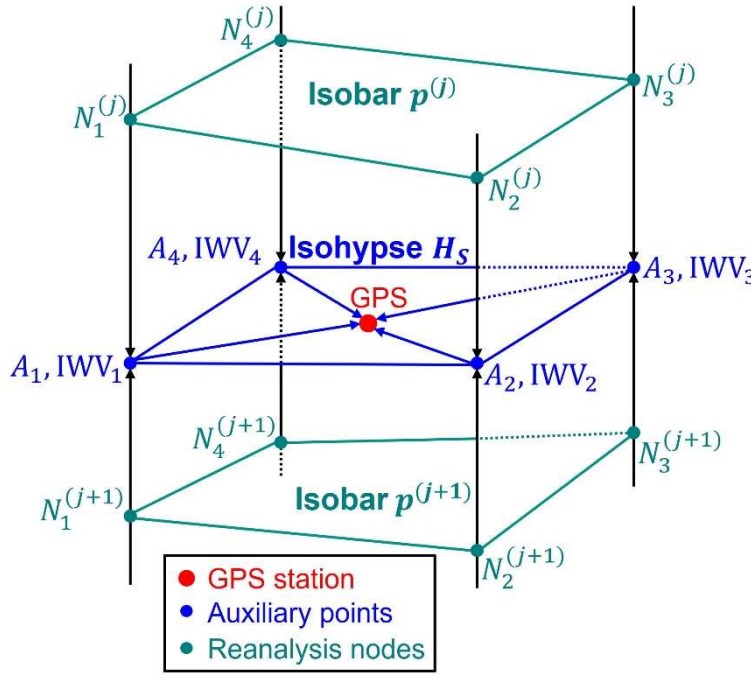

**Figure 2.** Schematic plot of the vertical and horizontal interpolation of the reanalysis pressure level products. The IWV at each auxiliary point ($A_i$; orange dots) is calculated with vertical interpolation or extrapolation of the adjacent reanalysis nodes ($N_i^{(j)}$ and $N_i^{(j+1)}$; green dots). The IWV at the GPS station (red dot) is then estimated with horizontal interpolation of the auxiliary points.

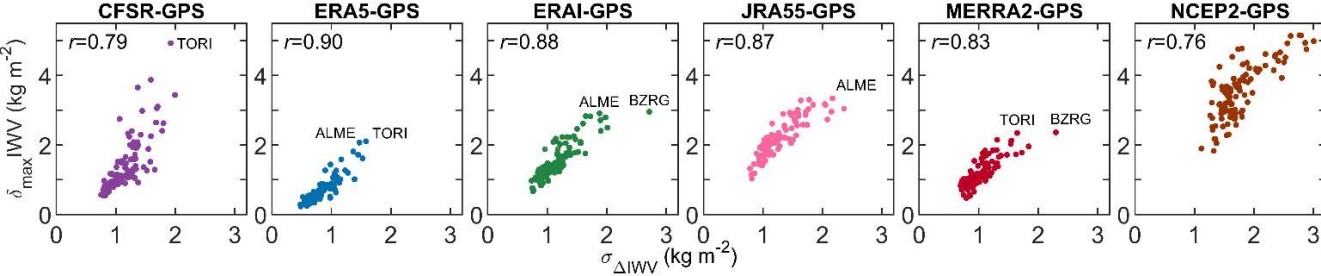

**Figure 3.** Scatterplots showing the relationships between the standard deviations of daily IWV difference ($\sigma_\Delta$) and the representativeness error statistic ($\delta_{max}$) for the 108 GPS stations.

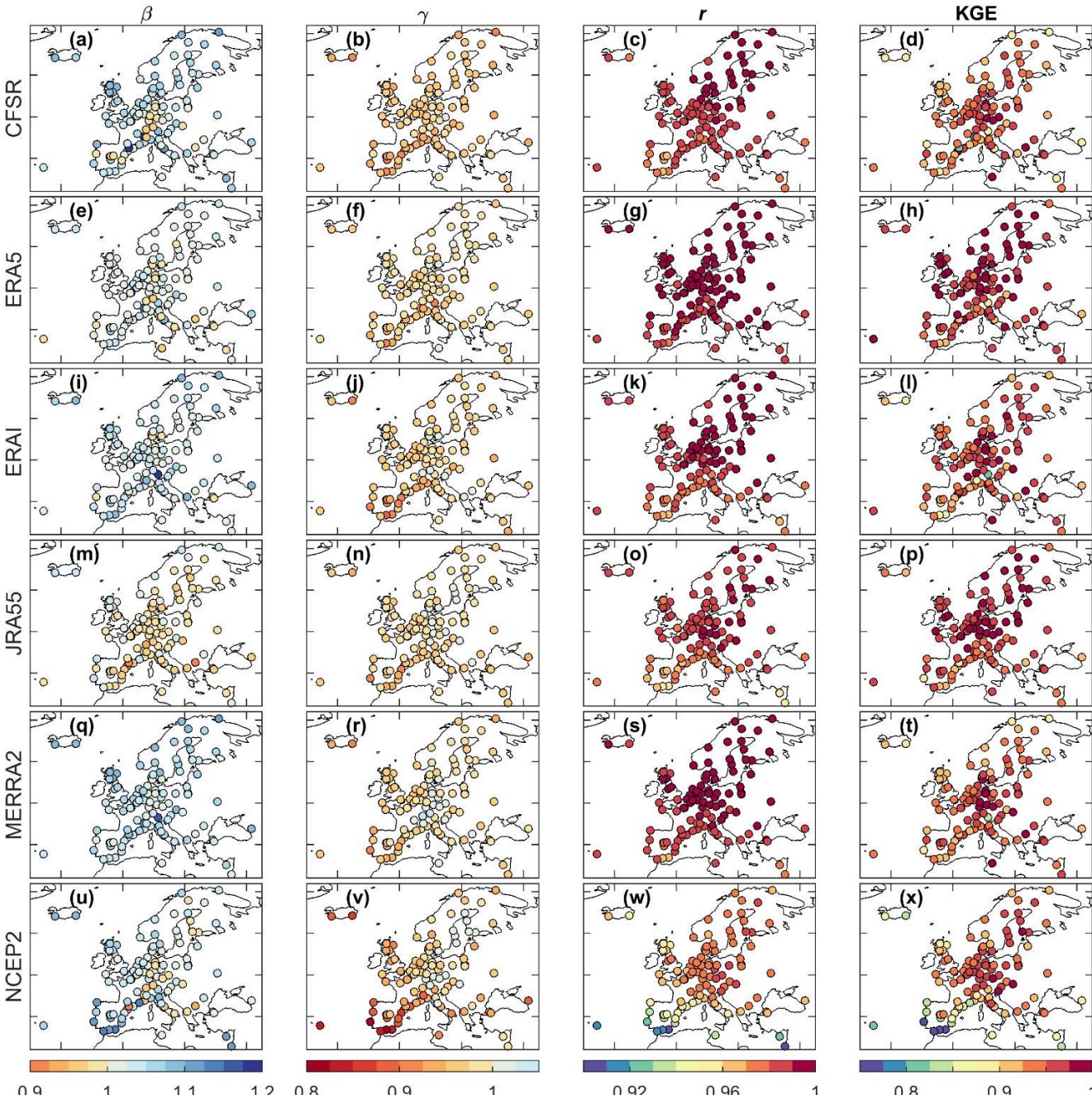

**Figure 4.** Plots of the KGE parameters for the daily IWV time series of the 108 GPS stations. The $\beta$ and $\gamma$ indicate the consistencies in the mean and variability, respectively. The $r$ indicates the Pearson correlation coefficient between the GPS and reanalyses time series. With perfect consistencies in mean, variability, and correlation, the $\beta$, $\gamma$, and $r$ are identical to 1, respectively. When the $\beta$, $\gamma$, and $r$ are identical to 1, the KGE score will reach its maximum value of 1.

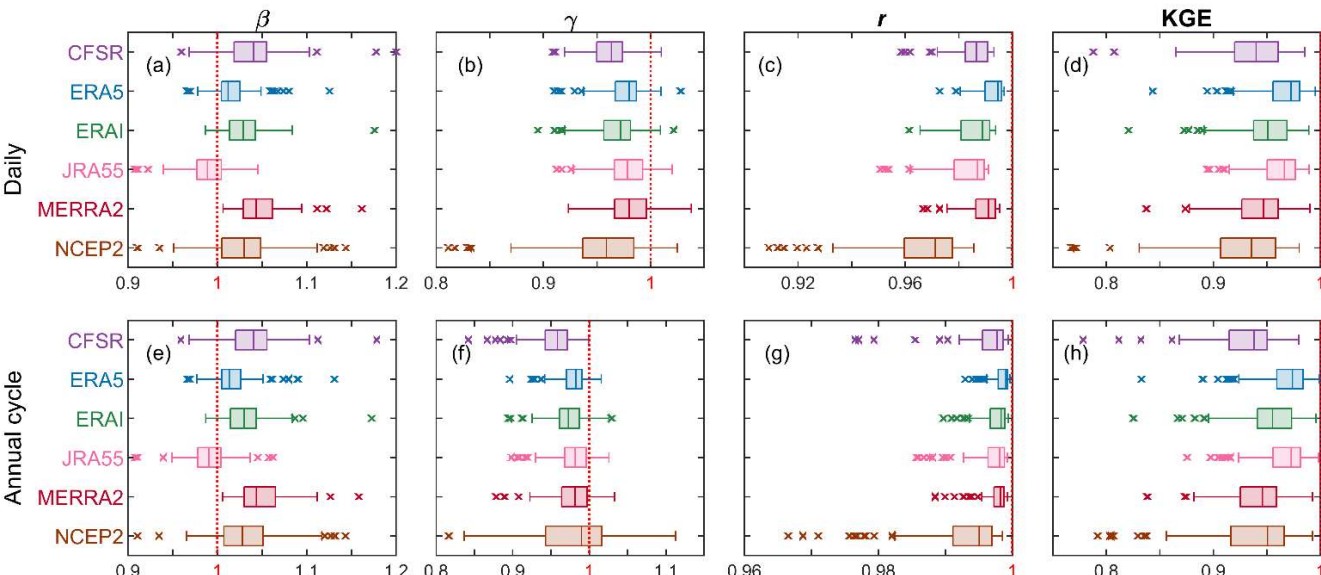

**Figure 5.** Box-whisker plots of the KGE parameters for the daily time series (a−d) and monthly annual cycle of IWV (e−h) from the reanalyses compared to GPS for the 108 stations.

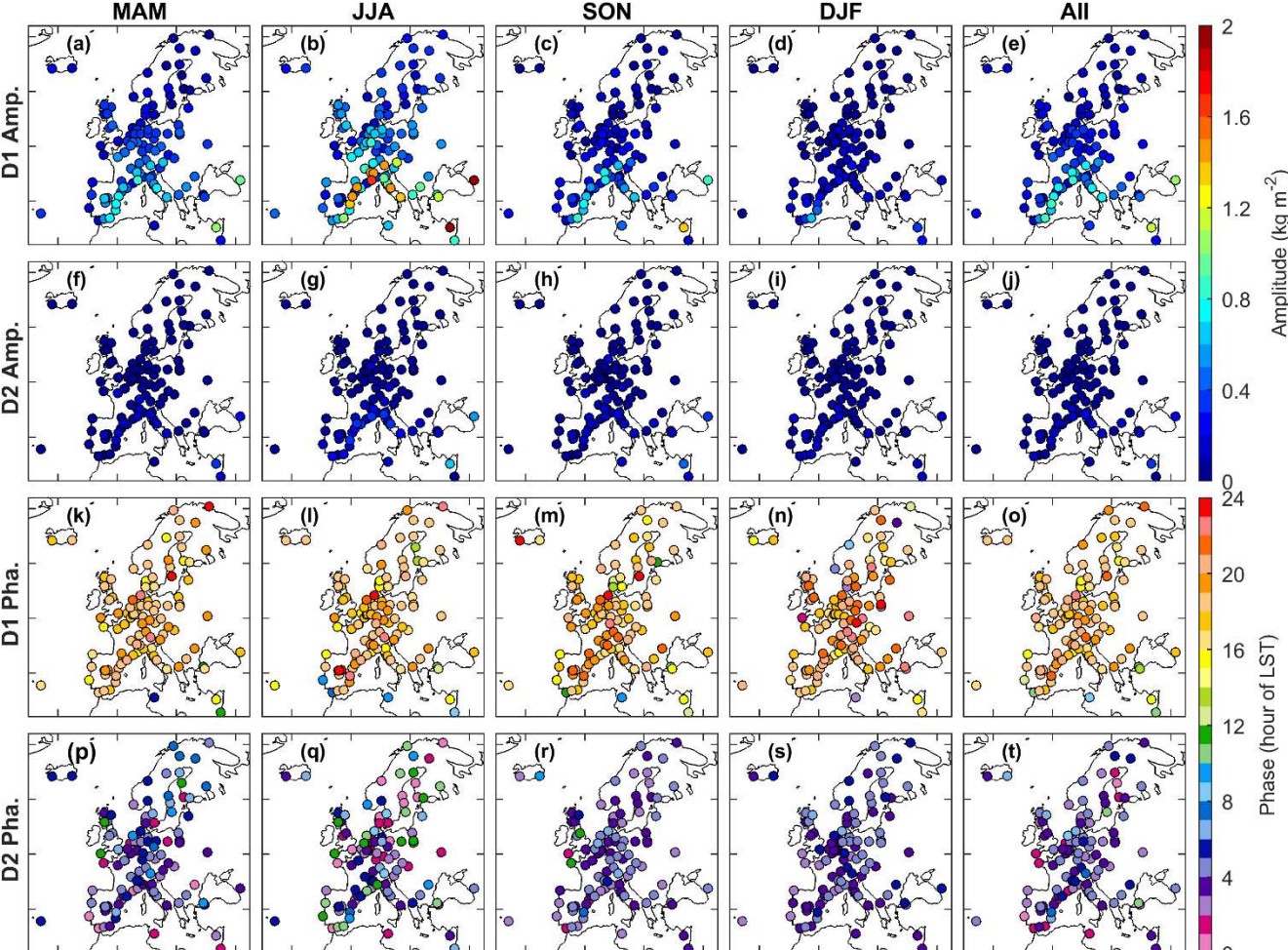

**Figure 6.** Plots of the amplitudes (a and f) and phases (k and p) for the first (D1) and second (D2) harmonics of diurnal GPS IWV cycle averaged in MAM (spring) at each station, respectively. The other subplots are for JJA (summer), SON (autumn), DJF (winter), and annual, from left to right, respectively. The phases are in Local Solar Time (LST) at the peak of associated harmonics.

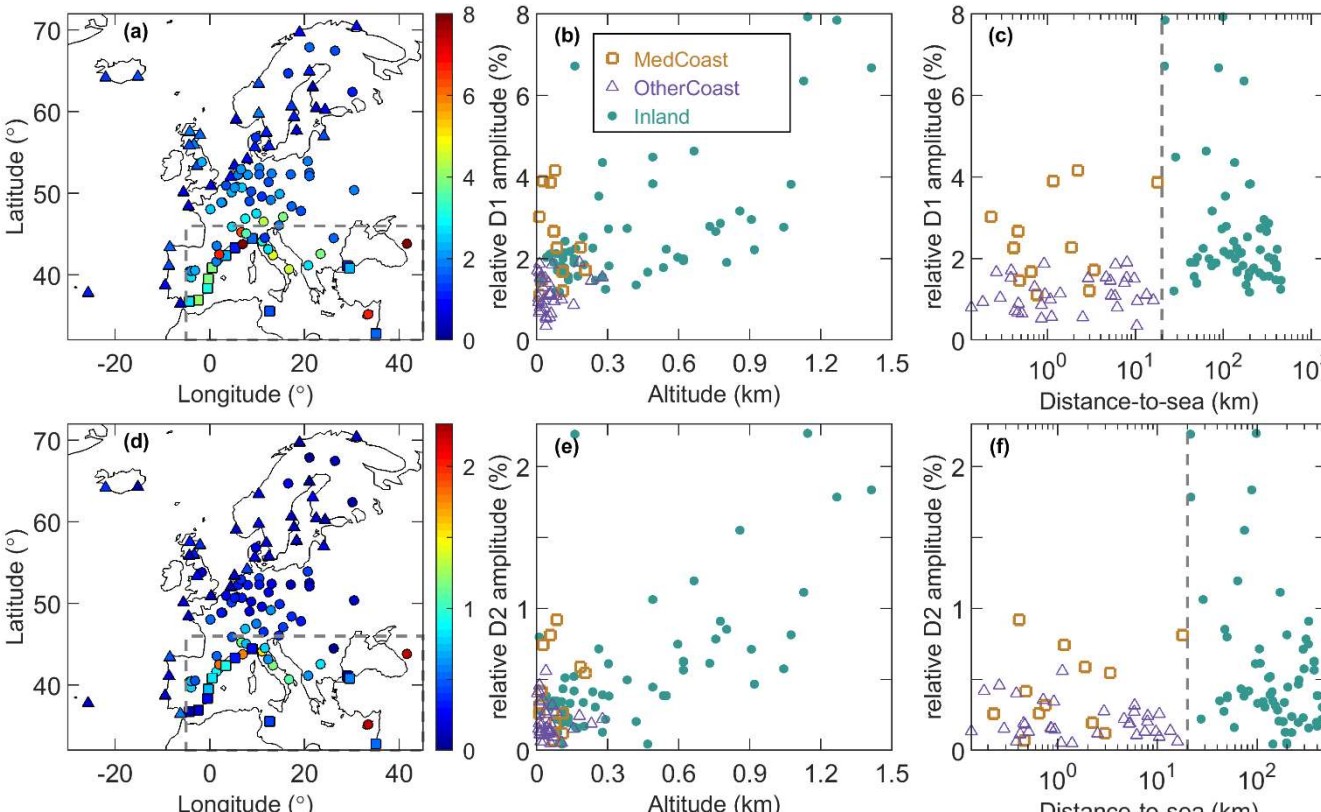

**Figure 7.** (a) relative amplitudes of the first harmonic ($D_1$) of diurnal GPS IWV cycle. (b) and (c) are the variations of the relative $D_1$ amplitudes with respect to station altitude and distance-to-sea, respectively. (d−f) are the same as (a−c) but for the second harmonic ($D_2$). The stations are classified into three types, namely Inland, MedCoast and OtherCoast. The type of Inland includes 62 stations with their distance to sea (SeaDist) no shorter than 20 km. The type of MedCoast contains 12 stations located at the coastal region of Mediterranean (SeaDist<20 km, 32 °N <Lat<46 °N, 5 °W<Lon<45 °E). The rest 34 stations are classified as OtherCoast.

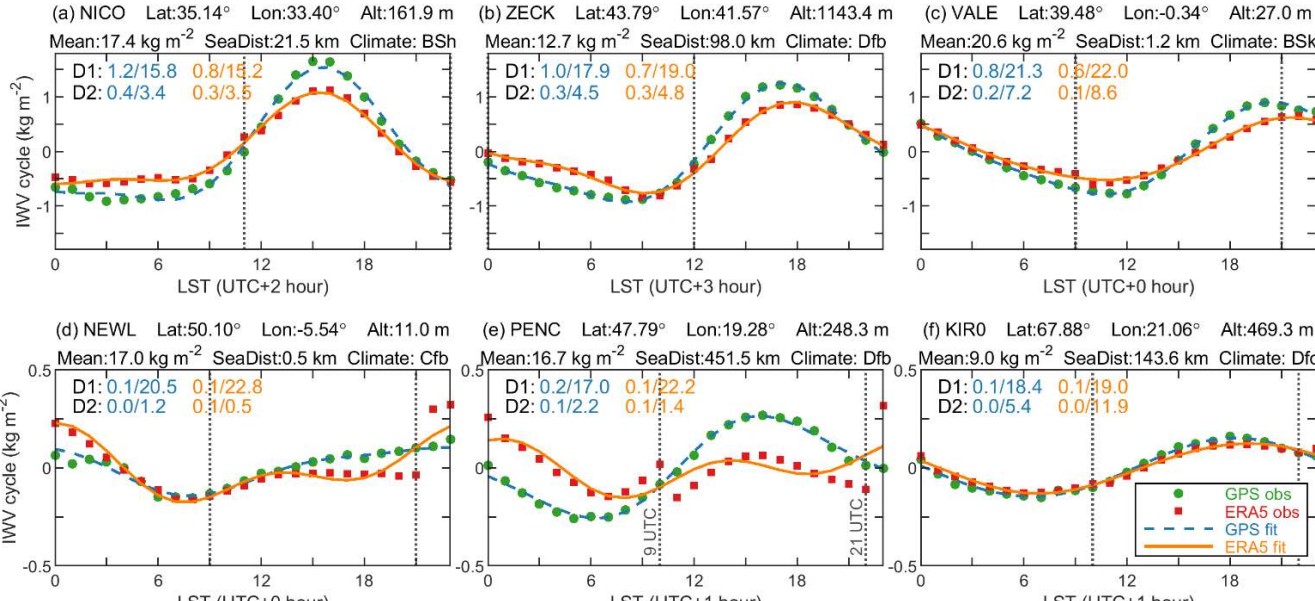

**Figure 8.** Diurnal IWV cycles at selected six stations obtained from 1-hourly GPS (green dots) and ERA5 (red squares). The stations are selected with the consideration of different altitudes, distance-to-sea (SeaDist), and climate zones classified according to Köppen Climate Classification (Beck et al., 2018). The data points are fitted with diurnal (D1) and semidiurnal (D2) harmonics (blue dashed curve for GPS and orange curve for ERA5). The amplitudes and phases of the D1 and D2 harmonics are also given. The phases are shown as the Local Solar Time (LST) at the peak of associated harmonics. For instance, the D1 amplitude and phase of GPS IWV at station NICO are 1.2 kg m$^{-2}$ and 15.8 LST, respectively. By comparison, the values of its ERA5 IWV are 0.8 kg m$^{-2}$ and 15.2 LST, respectively. The vertical black dotted lines at 09 and 12 UTC indicate the time of possible mismatches in the ERA5 IWV cycle.

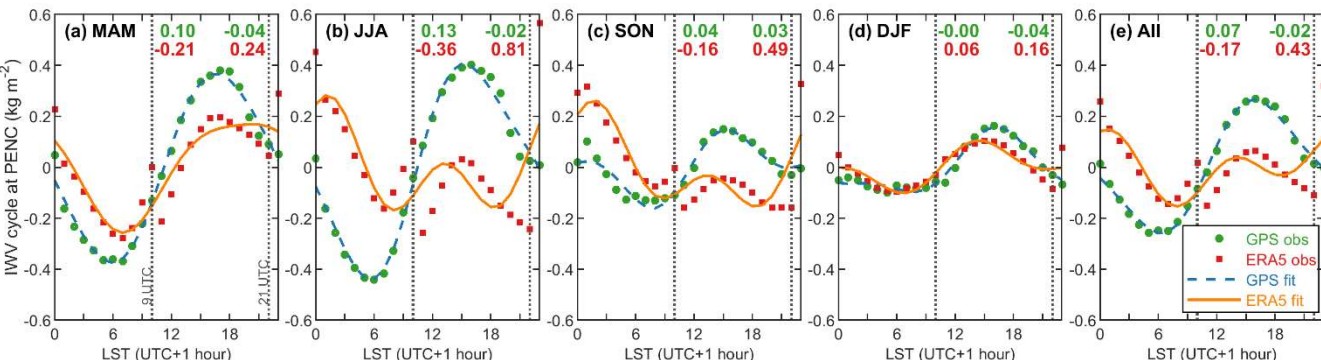

**Figure 9.** (a) Similar to Fig. 8 but for the seasonal and all-time averaged diurnal IWV cycles at station PENC. The green and red numbers are the IWV shifts from ERA5 and GPS, respectively. The numbers on the left and right are the IWV shifts from 09 to 10 UTC and from 21 to 22 UTC, respectively.

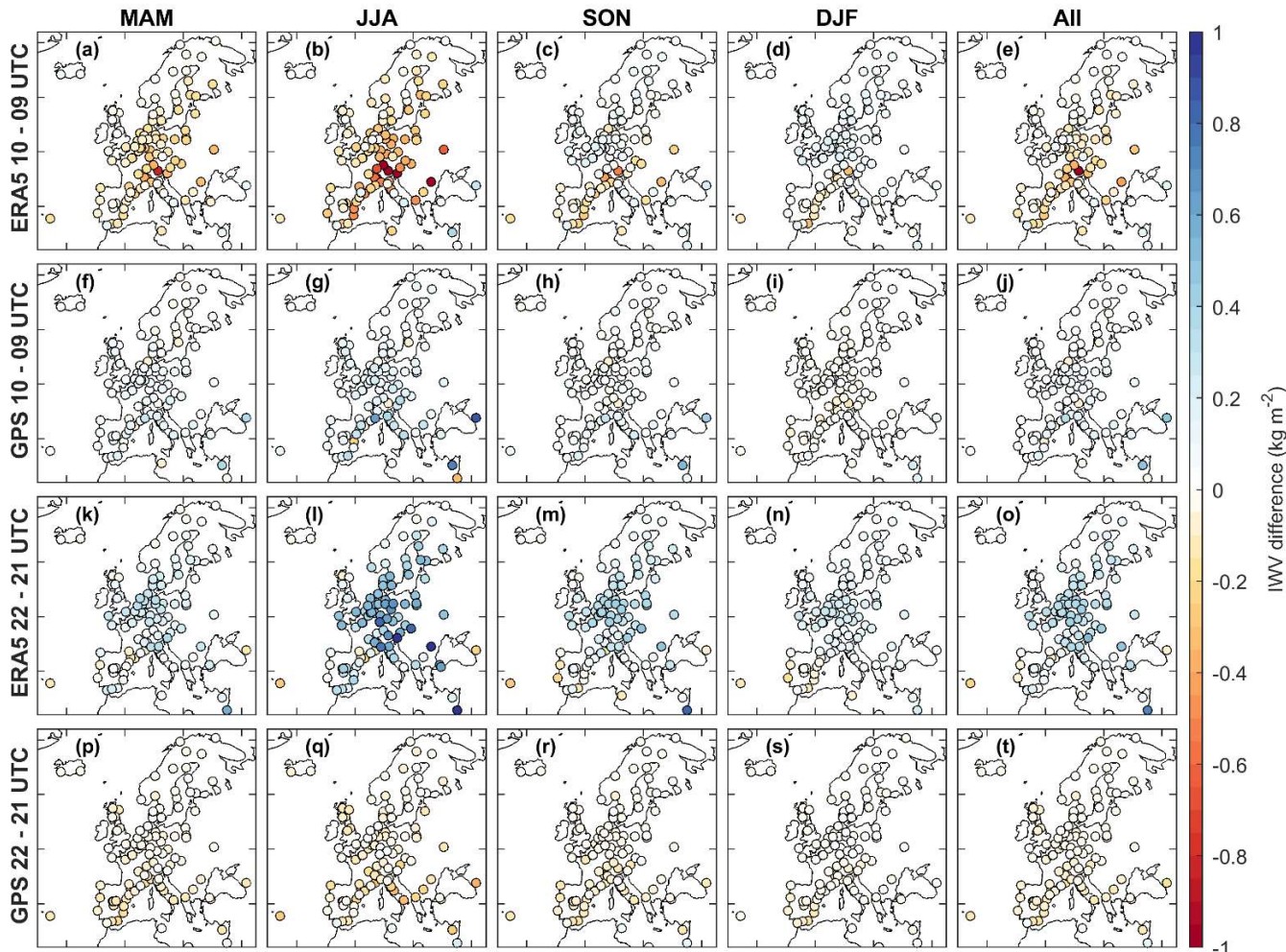

**Figure 10.** Seasonal and all-time averaged IWV shifts in the GPS and ERA5 diurnal cycles from 09 to 10 UTC and from 21 to 22 UTC.

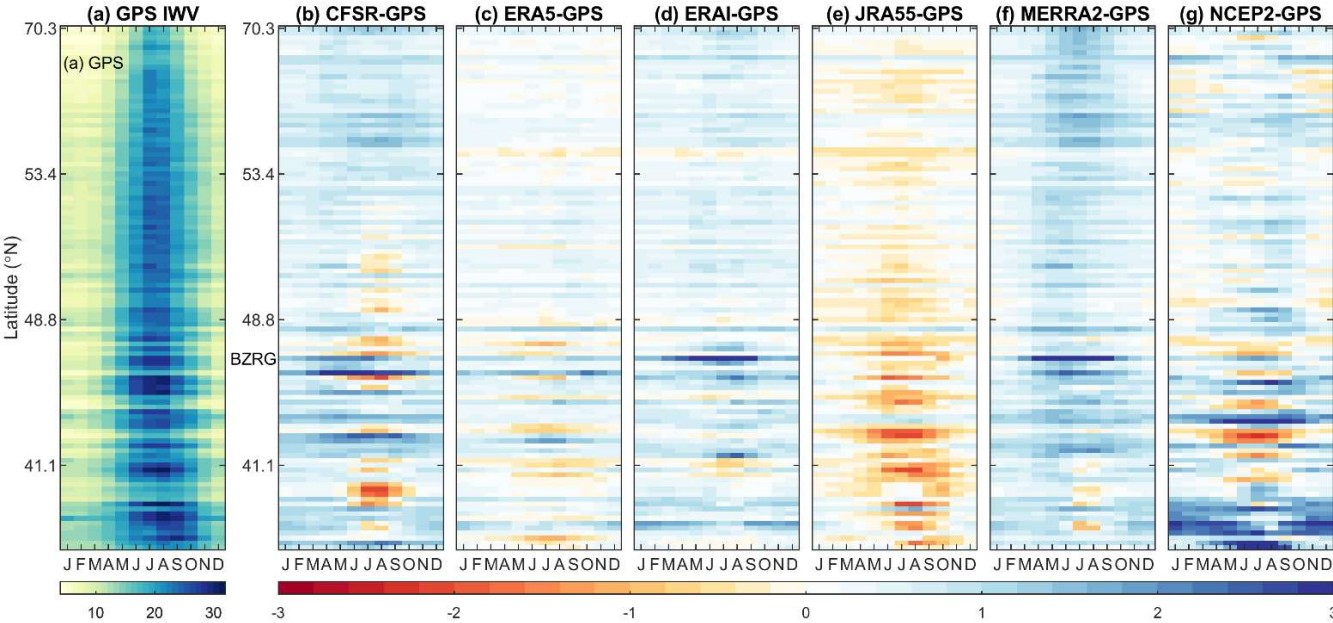

**Figure 11.** (a) Monthly annual cycle of GPS IWV in kg m$^{-2}$ for the 108 GPS stations with increasing latitude from bottom up. (b−g) are the differences of annual cycle from the various reanalyses compared to the GPS.

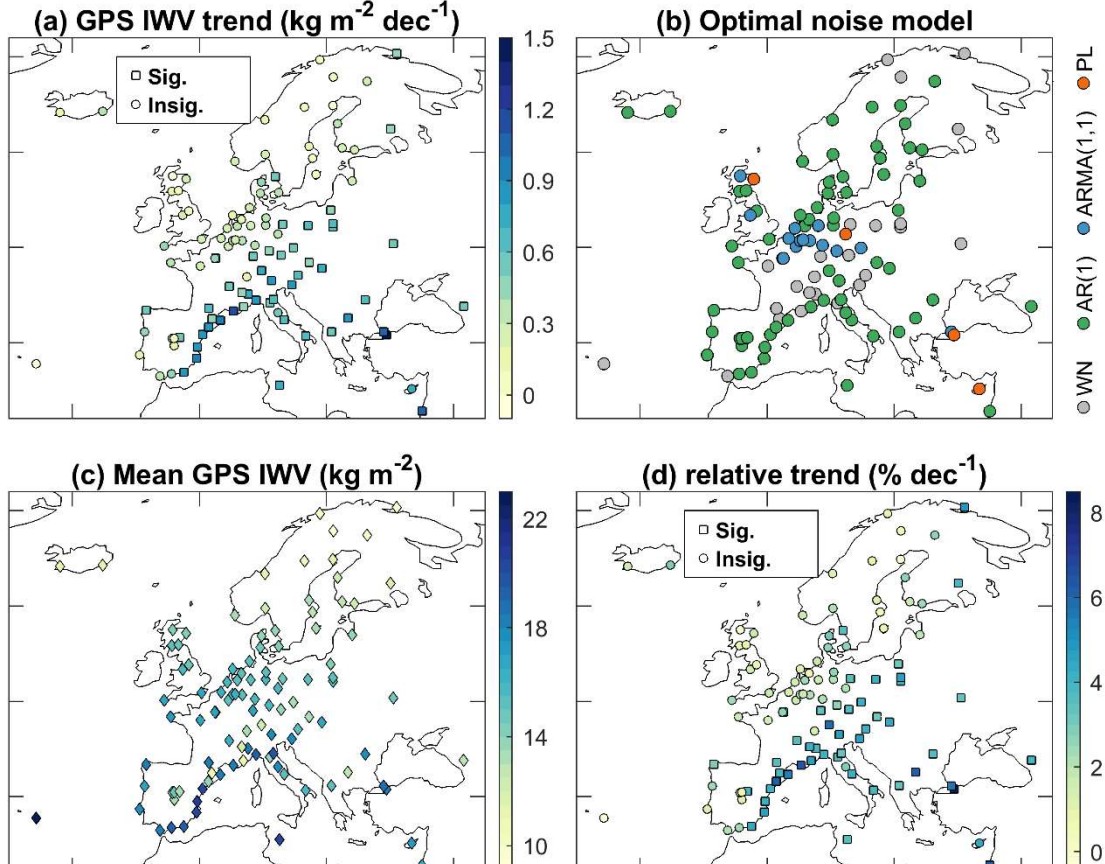

**Figure 12.** (a) Map of the absolute trends of the homogenised monthly GPS IWV anomaly time series. The squares and circles indicate the trend estimates being significant and insignificant at 95% confidence level, respectively. The trend uncertainties are estimated based on optimal noise models (b). The relative trends (d) are calculated as the absolute trends (a) divided by the associated average IWV (c).

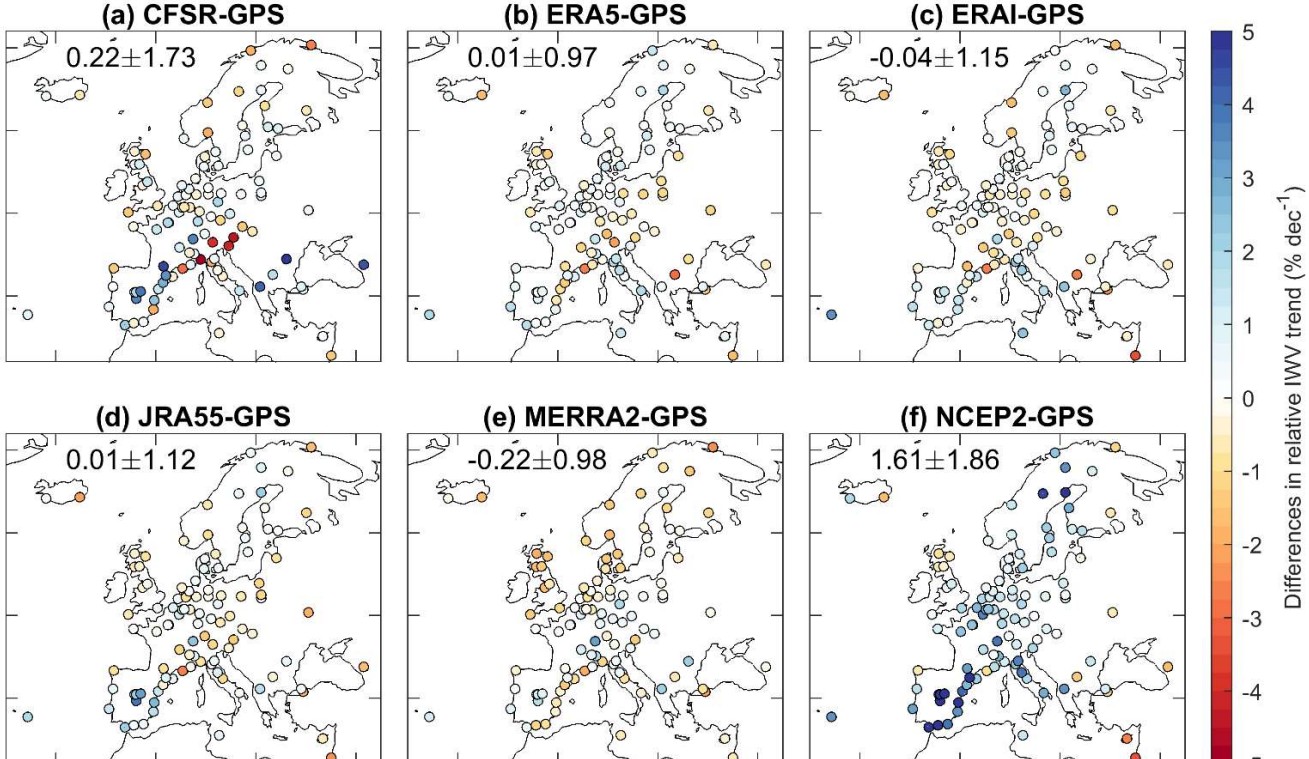

**Figure 13.** Comparison of the relative IWV trend differences (% decade$^{-1}$) for various reanalyses with respect to GPS, respectively. The numbers in each subplot indicate the mean value and standard deviation of the associated relative IWV trend differences, respectively.

**Table 1.** Six atmospheric reanalyses used in this study and their characteristics.

| Reanalysis | CFSR | ERA5 | ERAI | JRA55 | MERRA2 | NCEP2 |
|---|---|---|---|---|---|---|
| Source | NCEP | ECMWF | ECMWF | JMA | GMAO | NCEP |
| Assimilation | 3DVAR | 4DVAR | 4DVAR | 4DVAR | 3DVAR | 3DVAR |
| Time range | 1979−present | 1979−present | 1979−Aug. 2019 | 1958−present | 1980−present | 1979−present |
| Temporal res. (hour) | 6 | 1 | 6 | 6 | 3 | 6 |
| Horizontal res. (lat.×lon.) | 0.5°×0.5° | 0.25°×0.25° | 0.75°×0.75° | 1.25°×1.25° | 0.5°×0.625° | 2.5°×2.5° |
| Pressure levels | 37 | 37 | 37 | 37 | 42 | 17 |
| References | Saha et al., 2010, 2014 | Hersbach et al., 2020 | Dee et al., 2011 | Kobayashi et al., 2015 | Gelaro et al., 2017 | Kanamitsu et al., 2002 |

**Table 2.** Statistics of the consistencies in diurnal IWV anomalies from reanalyses compared to GPS.

| | Temporal resolution | CFSR | ERA5 | ERAI | JRA55 | MERRA2 | NCEP2 |
|---|---|---|---|---|---|---|---|
| $\gamma$ | 1 h | 0.98±0.04 | 0.98±0.03 | 0.91±0.04 | 0.86±0.04 | 1.02±0.04 | 0.75±0.05 |
| | 3 h | – | 0.97±0.03 | – | – | 1.02±0.04 | – |
| | 6 h | 0.99±0.03 | 0.96±0.03 | 0.92±0.04 | 0.87±0.04 | – | 0.76±0.05 |
| $r$ | 1 h | 0.85±0.05 | 0.89±0.05 | 0.85±0.06 | 0.83±0.06 | 0.86±0.06 | 0.69±0.08 |
| | 3 h | – | 0.89±0.05 | – | – | 0.86±0.06 | – |
| | 6 h | 0.87±0.05 | 0.90±0.06 | 0.86±0.07 | 0.84±0.07 | – | 0.69±0.08 |
| KGE | 1 h | 0.85±0.05 | 0.88±0.06 | 0.82±0.07 | 0.78±0.07 | 0.85±0.06 | 0.60±0.09 |
| | 3 h | – | 0.89±0.06 | – | – | 0.86±0.06 | – |
| | 6 h | 0.86±0.06 | 0.89±0.06 | 0.83±0.07 | 0.79±0.07 | – | 0.60±0.08 |
| $RMS_\Delta$ (kg m$^{-2}$) | 1 h | 1.14±0.19 | 0.97±0.21 | 1.14±0.22 | 1.20±0.21 | 1.14±0.22 | 1.57±0.24 |
| | 3 h | – | 0.97±0.21 | – | – | 1.15±0.22 | – |
| | 6 h | 1.09±0.21 | 0.94±0.21 | 1.09±0.24 | 1.16±0.22 | – | 1.59±0.24 |

**Appendix A: Homogenisation of GNSS IWV time series**

Long-term IWV time series often suffer from inhomogeneities due to changes in instrumentation, data processing methods, and local environmental conditions (Van Malderen et al., 2020; Nguyen et al., 2021 and references therein). These inhomogeneities can manifest themselves as changes in the mean of the time series ("biases") at specific epochs, i.e., breaks or changepoints. If such changepoints are not properly corrected, they can significantly modify the estimations of long-term linear trend and multi-temporal-scale variabilities (e.g., Ning et al., 2016; Van Malderen et al., 2020; Yuan et al., 2021). Therefore, the homogenisation of the IWV time series is essential for a sound understanding and proper interpretation of IWV variability under climate change.

In this work, we examined the homogeneity of the monthly GPS IWV time series by using the RHtestsV4 software (Wang and Feng, 2013). This software is developed especially for the detection and adjustment of changepoints in climatic time series, and it has been used in the homogenisation of IWV time series in previous studies (Ning et al., 2016; Schröder et al., 2016; Van Malderen et al., 2020). The software is based on a penalized maximal t test with the consideration of linear trend, annual cycle, and AR(1) noise in the time series (Wang et al., 2007; Wang 2008).

We took all the six reanalyses as references for the homogenisation of the GNSS IWV time series, meaning that we inspected the monthly IWV difference time series between the GNSS and each of the reanalysis IWV. It is noteworthy that the reanalyses may also contain changepoints (e.g., Ning et al., 2016; Schröder et al, 2016). However, we did not homogenise the reanalyses IWV time series because they represent the native quality of the reanalyses that we would like to assess in this work. Also, by taking all six reanalyses as references, we are confident to minimise the impact of inhomogeneities in either reanalysis on the homogenisation process of the GNSS IWV time series. Practically, we used the following strategy to avoid the impacts of changepoints in specific reanalyses on the homogenisation of the GPS IWV time series:

(1) We examined the GPS IWV and metadata (station log file and IGSMAIL) carefully. If there is an instrumentation change within the first (or the last) year, we removed the several months before (or after) the epoch of change. Moreover, we also inspected the station up-coordinate time series and excluded periods with quality problem, as it is well-known that they are strongly correlated with the GPS tropospheric delay estimates (Tregoning and Herring, 2006). An example is given in Fig. A1 and will be described later in this appendix.

(2) We used the *FindU.wRef* command of the RHtestsV4 software to identify all possible changepoints in each GPS-reanalysis IWV monthly mean difference time series, which can be significant at a confidence level of 99% no matter they are documented in metadata or not.

(3) If a changepoint is within three months before or after a documented change in instrumentation, we adjusted its epoch according to the metadata and set it as Type-0. The rest changepoints are set as Type-1.

(4) If identical Type-1 changepoints are reported within six months in at least four GPS-reanalysis IWV differences, but not supported by metadata, we recognised them as a single Type-1 changepoint at the median of the epochs.

(5) We estimated the amplitudes of the Type-0 and Type-1 changepoints and tested their amplitude significances at a confidence level of 99% with the *StepSize.wRef* command of the RHtestsV4 software.

(6) We calculated the amplitude of each changepoint as the average of all the significant amplitude estimates from the six GPS-reanalysis IWV differences.

(7) We removed a changepoint if its amplitude is only significant in less than four GPS-reanalysis comparisons or its amplitude is less than three times of its standard deviation. Then, we repeated the steps of (5) and (6) until all the rest changepoints are significant.

The changepoint identification is finished after one or two iterations for most stations. In the end, we identified 44 Type-0 and 9 Type-1 changepoints as listed in Table A1. The total number of 53 changepoints is consistent to a previous global GPS IWV homogenisation work carried out by Ning et al. (2016), which identified 45 changepoints in total at 101 stations.

As the changepoint detection was carried out on monthly level, the specific dates of the Type-0 changepoint 885 are fixed as documented. However, for the Type-1 changepoints without supports from metadata, their time of occurrences are fixed to the 15th day of associated month. We adjusted the GPS IWV time series by adding the amplitude of each changepoint to the GPS IWV data points before its time of occurrence. Note that five out of the nine Type-1 changepoints are significant in all the six GPS-reanalyses comparisons. Although we tried to minimize the impacts of changepoints in specific reanalyses on the results here, we cannot completely rule out 890 that identical changepoints appear in all the six reanalyses by ingesting the same observational datasets through data assimilation.

Figure A1 and A2 show the homogenisation results at two stations. Station HERS (Herstmonceux, UK, 0.34 °E, 50.87 °N) is characterised with abnormal variations in its up-coordinate time series before the changes of antenna and receiver on 1998-2-18 as shown in Fig. A1h, indicating low-quality observations related to the 895 instrumentation. In addition, obvious abnormal variations can be seen in all the GPS-reanalysis comparisons before 2001 September. We checked IGSMAIL-3503 (https://lists.igs.org/pipermail/igsmail/2001/004876.html) which reported a repair of antenna at station HERS until 2001-9-3. Therefore, we excluded the GPS IWV data before the date of repair. Then, we used the RHtestsV4 software to identify the changepoints in the rest GPS

IWV time series and found one on 2010-8-19, which is significant at a confidence level of 99%. It is a Type-0

changepoint due to an antenna and receiver changes as recorded in the station log file. After the homogenisation, the linear trend of the GPS IWV time series at station HERS has been reduced from 0.71 to 0.27 kg m$^{-2}$ decade$^{-1}$, which generally agrees better with the trend estimates from reanalyses, which are 0.15, 0.43, 0.33, 0.37, 0.10, and 0.59 kg m$^{-2}$ decade$^{-1}$ for CFSR, ERA5, ERAI, JRA55, MERRA2, and NCEP2, respectively.

Station ERLA (Erlangen, Germany, 11.01 °E, 49.59 °N) has a Type-1 changepoint in 2015 July for

unknown reason, in addition to a Type-0 one due to antenna and radome changes on 2010-8-18 (Fig. A2). The date of the Type-1 changepoint was fixed to 2005-7-15. With the homogenisation, the linear trend of the GPS IWV time series at station ERLA has been increased from 0.14 to 0.40 kg m$^{-2}$ decade$^{-1}$, which is closer to the trend estimates from reanalyses, which are 0.43, 0.36, 0.41, 0.43, 0.52, and 0.68 kg m$^{-2}$ decade$^{-1}$ for CFSR, ERA5, ERAI, JRA55, MERRA2, and NCEP2, respectively. Moreover, we compared the GPS IWV trend to three

nearby stations (KARL, KLOP, and WTZR) with values of 0.56, 0.32, and 0.66 kg m$^{-2}$ decade$^{-1}$ and their distances to ERLA of 198.6, 177.5, and 144.4 km, respectively. The results indicate an improved spatial consistency in the GPS IWV trends from the homogenised time series. Therefore, the homogenisation at station ERLA is considered to be reasonable.

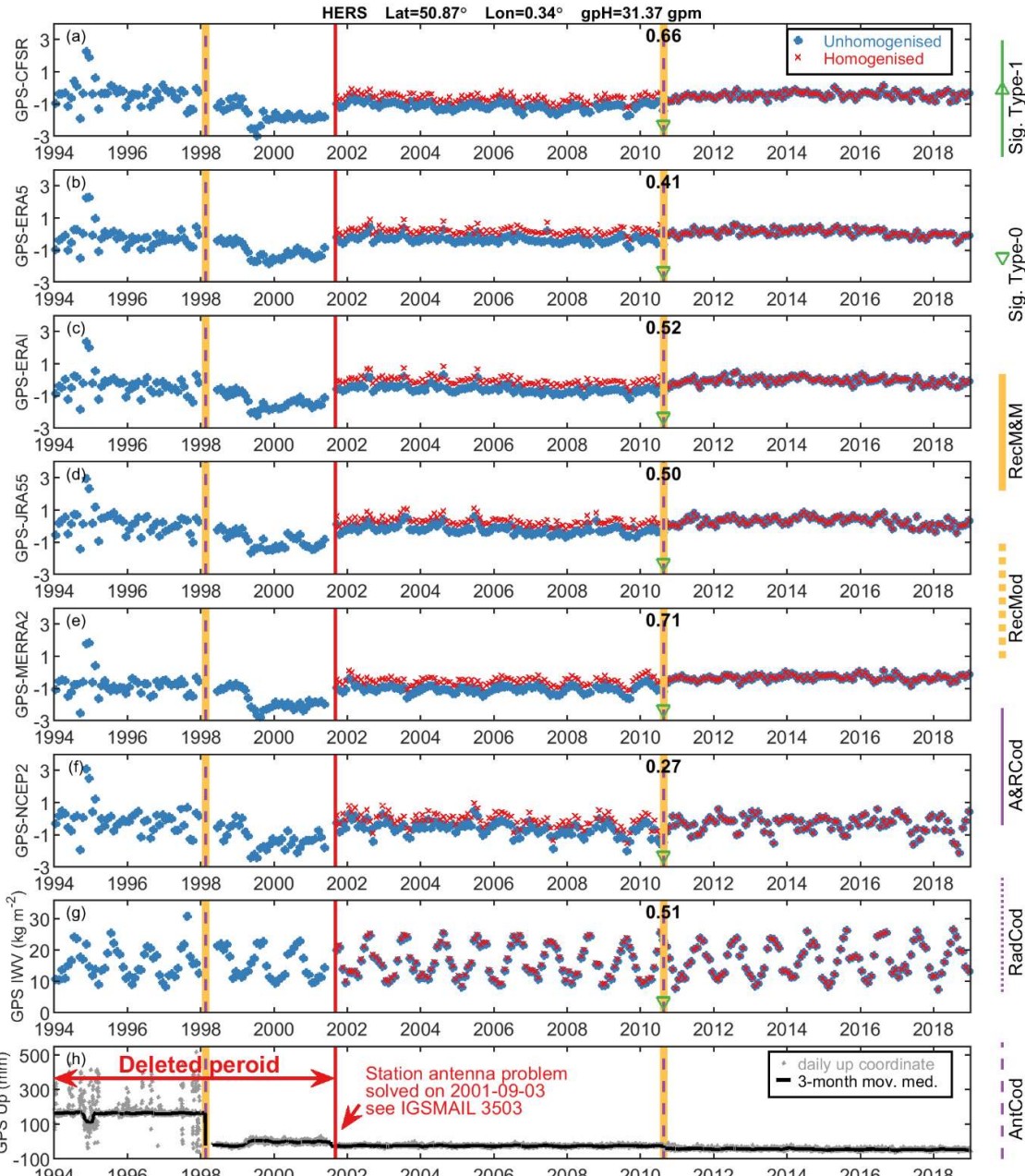

**Figure A1.** Monthly GPS-reanalysis IWV difference (a−f), monthly GPS IWV (g), and daily up-coordinate time series at station HERS. The GPS data before 2001-9-3 were deleted due to a problem in station antenna (see IGSMAIL 3503). The IWV time series before and after the homogenisation are labelled as blue dots and red crosses in (a)−(g). The types of instrumentation changes are listed in Table A2. The Type-0 changepoint on 2010-8-19 is significant at a confidence level of 99% (green downward-pointing triangle), and its value was calculated as the average of the values estimated in each GPS-reanalysis comparison as shown in (a−f).

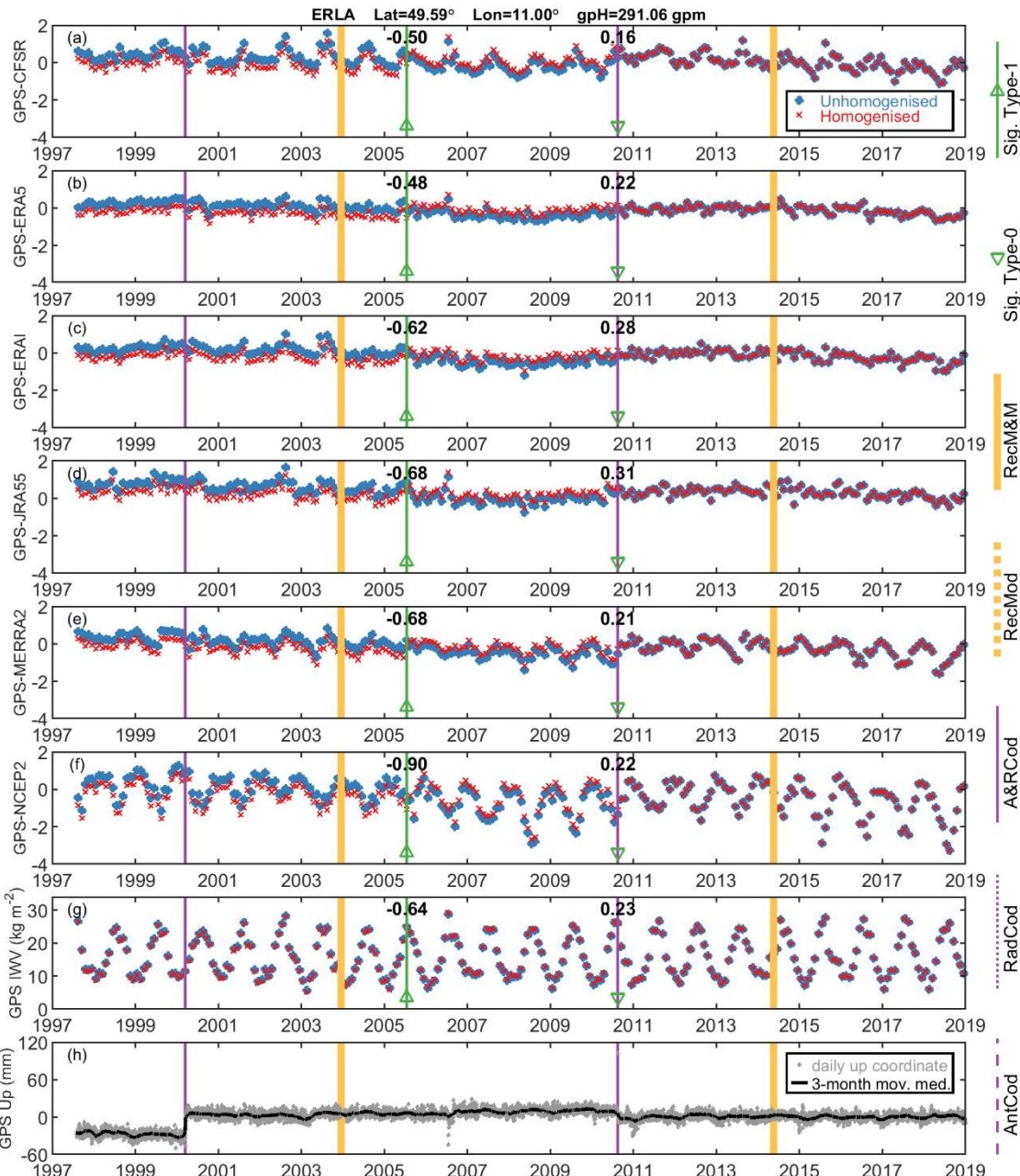

**Figure A2.** The same as Fig. A1 but for station ERLA. A Type-0 changepoint is significant on 2010-8-18 (green triangle) due
to changes in antenna and radome at the station. A Type-1 changepoint in 2005 July (green vertical line and upward-pointing
triangle) is significant but without support from metadata, and hence its date was fixed to 2005-7-15.

**Table A1.** The identified changepoints in GPS IWV time series. The Type-0 and Type-1 are changepoints with and without changes in instrumentation as documented in GPS station log files, respectively. The full names of the Type-0 changepoint events are shown in Table A2, and the Type-1 changepoints are labelled as Unknown. The G-C, G-E, G-I, G-J, G-M, and G-N indicate the GPS IWV changepoints in kg m$^{-2}$ estimated by compared to CFSR, ERA5, ERAI, JRA55, MERRA2, and NCEP2, respectively. The NaN indicates that the changepoint in the specific GPS-reanalysis comparison is insignificant at a confidence level of 99%. The mean and SD are the mean value and standard deviation of each changepoint.

| Station | Type | Date | Mean | SD | G-C | G-E | G-I | G-J | G-M | G-N | Events |
|---------|------|------|------|-----|-----|-----|-----|-----|-----|-----|--------|
| APEL | 1 | 2006-09-15 | -0.31 | 0.10 | -0.26 | -0.30 | -0.32 | -0.22 | -0.26 | -0.50 | Unknown |
| APEL | 0 | 2013-06-19 | 0.36 | 0.10 | 0.36 | 0.32 | 0.40 | 0.44 | 0.47 | 0.20 | RecM&M |
| APEL | 1 | 2016-04-15 | -0.33 | 0.04 | -0.37 | -0.29 | NaN | -0.32 | -0.33 | NaN | Unknown |
| BELL | 0 | 2012-01-26 | 1.22 | 0.30 | 0.67 | 1.43 | 1.30 | 1.34 | 1.47 | 1.11 | AntCod |
| BELL | 0 | 2014-09-22 | -0.47 | 0.13 | -0.36 | -0.49 | -0.49 | -0.32 | -0.46 | -0.69 | AntCod |
| BELL | 0 | 2016-05-12 | -1.61 | 0.05 | -1.56 | -1.57 | -1.59 | -1.71 | -1.63 | -1.63 | AntCod and RecM&M |
| BRST | 0 | 2011-10-26 | 0.36 | 0.08 | 0.45 | 0.28 | 0.39 | 0.41 | 0.37 | 0.25 | AntCod and RecM&M |
| CREU | 0 | 2016-05-17 | -1.03 | 0.17 | -0.81 | -1.11 | -1.08 | -1.04 | -0.87 | -1.28 | AntCod and RecM&M |
| DELF | 0 | 2000-07-23 | 0.23 | 0.05 | NaN | 0.26 | 0.27 | 0.20 | 0.18 | NaN | A&RCod and RecM&M |
| DOUR | 0 | 2015-03-02 | 0.37 | 0.03 | 0.40 | 0.35 | 0.35 | 0.34 | 0.39 | NaN | AntCod |
| EIJS | 0 | 2000-04-28 | 0.22 | 0.07 | 0.33 | 0.25 | 0.19 | 0.18 | 0.15 | NaN | A&RCod and RecM&M |
| ERLA | 1 | 2005-07-15 | -0.64 | 0.15 | -0.50 | -0.48 | -0.62 | -0.68 | -0.68 | -0.90 | Unknown |
| ERLA | 0 | 2010-08-18 | 0.23 | 0.05 | 0.16 | 0.22 | 0.28 | 0.31 | 0.21 | 0.22 | A&RCod |
| EUSK | 0 | 2001-05-09 | -0.74 | 0.13 | -0.56 | -0.74 | -0.76 | -0.73 | -0.71 | -0.97 | A&RCod |
| GOPE | 0 | 1999-11-04 | 0.27 | 0.03 | 0.25 | 0.24 | NaN | NaN | 0.31 | 0.28 | A&RCod and RecM&M |
| GOPE | 0 | 2000-07-24 | -0.96 | 0.07 | -1.06 | -0.86 | -0.97 | -1.00 | -0.98 | -0.92 | A&RCod and RecM&M |
| GOPE | 1 | 2001-09-15 | -0.51 | 0.10 | -0.52 | -0.57 | -0.34 | -0.51 | -0.65 | -0.49 | Unknown |
| GOPE | 0 | 2006-07-14 | -0.39 | 0.09 | -0.39 | -0.30 | -0.32 | -0.33 | -0.47 | -0.52 | A&RCod |
| GOPE | 0 | 2009-12-14 | 0.35 | 0.10 | 0.17 | 0.33 | 0.40 | 0.43 | 0.33 | 0.42 | A&RCod and RecM&M |
| GOPE | 1 | 2016-05-15 | -0.36 | 0.06 | NaN | -0.29 | -0.36 | -0.33 | -0.35 | -0.46 | Unknown |
| HELG | 0 | 2008-09-02 | 0.19 | 0.06 | 0.23 | 0.15 | 0.11 | 0.19 | 0.27 | 0.18 | A&RCod and RecM&M |
| HELG | 0 | 2014-09-09 | -0.14 | 0.04 | NaN | -0.19 | -0.10 | -0.11 | NaN | -0.17 | A&RCod and RecM&M |
| HERS | 0 | 2010-08-19 | 0.51 | 0.16 | 0.66 | 0.41 | 0.52 | 0.50 | 0.71 | 0.27 | AntCod and RecM&M |
| HOBU | 0 | 2007-02-28 | -0.31 | 0.08 | -0.23 | -0.24 | -0.33 | -0.40 | -0.24 | -0.40 | A&RCod and RecM&M |
| HOBU | 0 | 2010-11-22 | 0.40 | 0.08 | 0.34 | 0.28 | 0.44 | 0.50 | 0.40 | 0.45 | A&RCod |
| HOBU | 0 | 2015-05-27 | -0.22 | 0.06 | -0.27 | -0.19 | -0.20 | -0.13 | -0.21 | -0.30 | RecM&M |
| HOFN | 0 | 2001-09-21 | -0.95 | 0.09 | -1.11 | -0.91 | -0.92 | -0.85 | -0.98 | -0.91 | A&RCod |
| KARL | 0 | 2001-05-10 | -0.64 | 0.14 | -0.64 | -0.47 | -0.54 | -0.65 | -0.68 | -0.89 | A&RCod |

| KLOP | 0 | 2001-05-08 | -0.60 | 0.11 | -0.40 | -0.55 | -0.65 | -0.67 | -0.62 | -0.72 | A&RCod |
| LAMP | 1 | 2013-03-15 | 0.99 | 0.17 | 1.05 | 0.70 | NaN | 1.00 | 1.10 | 1.10 | Unknown |
| LAMP | 0 | 2014-04-11 | -1.75 | 0.18 | -1.53 | -1.60 | -1.94 | -1.82 | -1.66 | -1.95 | AntCod and RecM&M |
| LAMP | 0 | 2017-09-26 | 0.52 | 0.13 | 0.48 | 0.36 | 0.69 | 0.47 | 0.59 | NaN | AntCod |
| LEED | 0 | 2008-11-11 | 0.38 | 0.11 | 0.24 | 0.38 | 0.39 | 0.46 | 0.53 | 0.29 | A&RCod |
| LEIJ | 0 | 2010-07-01 | 0.25 | 0.06 | NaN | 0.23 | 0.31 | 0.29 | NaN | 0.17 | A&RCod |
| MAN2 | 0 | 2008-01-23 | 0.29 | 0.04 | NaN | 0.24 | 0.32 | 0.32 | 0.28 | NaN | AntCod and RecMod |
| MODA | 0 | 2007-07-12 | 1.54 | 0.08 | NaN | 1.45 | 1.64 | 1.60 | 1.56 | 1.47 | RecM&M |
| MODA | 0 | 2008-01-08 | -1.60 | 0.28 | -1.11 | -1.53 | -1.69 | -1.68 | -1.92 | -1.70 | RecM&M |
| MOPI | 1 | 2004-08-15 | -0.53 | 0.09 | -0.43 | -0.54 | -0.51 | -0.44 | -0.65 | -0.62 | Unknown |
| OSNA | 0 | 2004-04-22 | 0.38 | 0.07 | 0.47 | 0.41 | 0.32 | 0.31 | 0.37 | NaN | AntCod |
| OSNA | 0 | 2007-04-23 | -0.61 | 0.06 | -0.60 | -0.57 | -0.62 | -0.59 | -0.57 | -0.73 | A&RCod and RecM&M |
| OSNA | 0 | 2011-04-05 | 0.23 | 0.06 | 0.18 | 0.14 | 0.28 | 0.29 | 0.24 | 0.26 | A&RCod |
| OSNA | 0 | 2015-06-11 | -0.18 | 0.06 | -0.17 | -0.20 | -0.12 | -0.16 | -0.18 | -0.28 | RecM&M |
| PENC | 0 | 2003-05-22 | 0.55 | 0.10 | 0.54 | 0.57 | 0.56 | 0.59 | 0.37 | 0.68 | AntCod |
| PENC | 0 | 2007-06-26 | -0.32 | 0.06 | -0.29 | -0.30 | -0.29 | -0.29 | -0.34 | -0.43 | RecM&M |
| PTBB | 1 | 2014-06-15 | -0.50 | 0.11 | -0.57 | -0.49 | -0.39 | -0.36 | -0.53 | -0.65 | Unknown |
| REYK | 1 | 2003-03-15 | 0.31 | 0.03 | NaN | 0.35 | 0.34 | 0.28 | 0.29 | NaN | Unknown |
| REYK | 0 | 2008-03-13 | -0.30 | 0.08 | -0.19 | -0.28 | -0.36 | -0.31 | -0.25 | -0.42 | A&RCod |
| REYK | 0 | 2013-05-02 | 0.27 | 0.09 | 0.24 | 0.13 | 0.34 | 0.33 | 0.28 | NaN | A&RCod and RecM&M |
| SULD | 0 | 2005-06-14 | -0.53 | 0.11 | -0.54 | -0.52 | -0.56 | -0.55 | -0.32 | -0.67 | AntCod |
| TERS | 0 | 2008-09-16 | -0.20 | 0.06 | -0.29 | -0.18 | -0.20 | -0.14 | -0.13 | -0.26 | RecM&M |
| TERS | 0 | 2013-08-29 | 0.35 | 0.05 | 0.30 | 0.37 | 0.39 | 0.36 | 0.41 | 0.28 | RecM&M |
| TRDS | 0 | 2007-05-07 | 0.25 | 0.06 | 0.27 | 0.20 | 0.27 | 0.18 | 0.32 | NaN | A&RCod |
| WSRA | 0 | 2000-01-06 | 0.14 | 0.02 | NaN | 0.15 | 0.16 | 0.11 | 0.13 | NaN | RecM&M |

935

**Table A2.** Types of changes in GPS instrumentation.

| | Abbreviation | Type of change |
|---|---|---|
| 1 | AntCod | Antenna Code |
| 2 | RadCod | Radome Code |
| 3 | A&RCod | Antenna and Radome Code |
| 4 | RecMod | Receiver Model |
| 5 | RecM&M | Receiver Make and Model |