# Peer review of "Characterisations of Europe's integrated water vapour and assessments of atmospheric reanalyses using more than two decades of ground-based GPS"

_Atmospheric Chemistry and Physics, 2021_

## Referee Comment (RC1)

**Manuscript Review**

**Title:** Characterizations of Europe's integrated water vapor and assessments of atmospheric reanalyses using more than two decades of ground-based GPS

**Authors:** Peng Yuan et al.

**Journal:** Atmospheric Chemistry and Physics

This manuscript investigated the multi-temporal-scale variabilities and trends of IWV and assessed six commonly-used atmospheric reanalyses (CFSR, ERA5, ERA-Interim, JRA55, MERR2, and NCEP2) over Europe using IWV time series from 108 GPS stations for more than two decades. I have the following comments:

**Main comments:**

1. The authors have taken into account of vertical IWV adjustment. However, the height system of GPS is different from that of the reanalyses. I'm not sure if the authors have considered the unification and differences of the different height systems?

2. In the manuscript, the authors used the difference time series between ERA5 IWV and GPS IWV to visually detect the breaks in GPS IWV, so the potential significant differences may be eliminated since the homogenization, also this may be the reason why the ERA5 outperforms than other reanalyses. Are these breaks based on ERA5 IWV still significant, are there any other reanalyses used for the homogenization process?

3. The spatial resolution contributes to most of the representativeness differences, such as the ERAI provides the products with higher spatial resolution (i.e. 0.25°) than the product used in this paper (0.75°). The conclusion that ERA5 has the best performance on the representativeness differences is questionable. This needs more clarification or convincing statements.

4. Line 202: "The 3- and 6-hourly IWVs are linearly interpolated into 1-hourly time series." Have the authors assessed the accuracy of the interpolated IWV? For IWV which changes in a high frequency, linear interpolation seems to be not a good choice.

5. Line 157: There seems to be a missing full stop between "reanalyses" and "Compared". Please check it.

---

## Community Comment (CC1)

Dear Peng Yuan and co-authors,

Thank you for releasing this interesting study. I am happy to see that you used the representativeness statistic that we proposed in a previous publication and that you confirm and extend our results to other reanalyses.

Below I submit a few questions and comments about your manuscript. Thank you in advance for your answers.

Best regards,

Olivier BOCK

1. Please comment on the choice and on the quality of the used GPS data set (NGL), as other data sets exist for Europe (e.g. the EPN repro2, Pacione et al., 2017).

2. Please provide more details on the homogenization method and results (e.g. the number and magnitude of detected breaks) and comment on their uncertainty. Explain also how the offsets in the GPS series are corrected, knowing that the breaks are detected in the GPS – reanalysis series and not in the GPS series directly.

   Regarding the homogenization method, I checked your earlier paper (Yuan et al., 2021), and was wondering why you used a manual segmentation method when many statistical methods exist, which have been assessed by Van Malderen et al., 2020. Can you comment on that choice?

   I also understand that in your segmentation method, you select only breaks which are confirmed by known equipment changes from the IGS log files. As you may have experienced: i) not all breaks are easy to detect (the example illustrated in Yuan et al., 2021, is a very optimistic case); ii) the IGS metadata may be incomplete and iii) the reanalysis may also have breaks. These limitations should be acknowledged in the paper.

   Moreover, regarding the first two points, I think the manual approach is very subjective and also probably too conservative. You mention in the former paper that you detected 21 breaks from 108 stations over 21 years, i.e. an average of 1 break per station every 108 years. This number is very small compared to other studies, e.g. Ning et al., 2016, and Nguyen et al., 2021, using statistical methods. Overall, Nguyen et al., 2021, detected 1 break per station every 5.8 years (after screening) considering all breaks, among which the validated cases represent 1 break per station every 16 years. Both studies also show some obvious examples of undocumented breaks (namely for HERS) and breaks attributed to the reanalysis. Regarding the last point, you write that no obvious breaks were found in the reanalysis. What are your criteria to detect breaks in the reanalysis?

3. The analysis of the diurnal cycle is interesting. However, to make a fair intercomparison, the reanalyses should be analysed at the smaller common resolution which is 6-hourly, and not interpolated to a higher resolution (1-hourly). For the two reanalyses which have higher resolution (ERA5 and MERRA-2), you may show both the native and under-sampled (6-hourly) results.

4. In section 3.2, you may mention that the moist bias of ERAI over Europe was also reported by Parracho et al. 2018.

5. Please explain how you compute the trends.

6. In Section 6, you may mention that the trend results are also in line with the findings of Parracho et al. 2018, and Nguyen et al., 2021.

7. What is MERRA2' in Figure 5?

Nguyen KN, Quarello A, Bock O, Lebarbier E. Sensitivity of Change-Point Detection and Trend Estimates to GNSS IWV Time Series Properties. *Atmosphere*. 2021; 12(9):1102. https://doi.org/10.3390/atmos12091102

Pacione, R., Araszkiewicz, A., Brockmann, E., and Dousa, J.: EPN-Repro2: A reference GNSS tropospheric data set over Europe, Atmos. Meas. Tech., 10, 1689–1705, https://doi.org/10.5194/amt-10-1689-2017, 2017.

---

## Author Comment (AC1)

**Community #1: https://doi.org/10.5194/acp-2021-797-CC1**

Dear Peng Yuan and co-authors,

Thank you for releasing this interesting study. I am happy to see that you used the representativeness statistic that we proposed in a previous publication and that you confirm and extend our results to other reanalyses.

Below I submit a few questions and comments about your manuscript. Thank you in advance for your answers.

Best regards,

Olivier BOCK

**Reply:** Thanks a lot for your comments.

**1.** Please comment on the choice and on the quality of the used GPS data set (NGL), as other data sets exist for Europe (e.g. the EPN repro2, Pacione et al., 2017).

**Reply:** Pacione et al. (2017) developed a GNSS tropospheric dataset of about 18 years (1996–2014) in the framework of EPN-Repro2. The authors also compared the results obtained with different GNSS data processing schemes. In this work, we selected the ZTD product developed by NGL as it has a much longer time series from 1994 Jan. to 2018 Dec. (25 years). As IWV is the meteorological parameter that we are interested in, we directly evaluated GPS IWV instead of ZTD. Our comparisons show that the IWV estimates from NGL's ZTD and various reanalyses are in good agreement, indicating that the ZTD product has good quality.

**2.** Please provide more details on the homogenization method and results (e.g. the number and magnitude of detected breaks) and comment on their uncertainty. Explain also how the offsets in the GPS series are corrected, knowing that the breaks are detected in the GPS – reanalysis series and not in the GPS series directly.

Regarding the homogenization method, I checked your earlier paper (Yuan et al., 2021), and was wondering why you used a manual segmentation method when many statistical methods exist, which have been assessed by Van Malderen et al., 2020. Can you comment on that choice?

I also understand that in your segmentation method, you select only breaks which are confirmed by known equipment changes from the IGS log files. As you may have experienced: i) not all breaks are easy to detect (the example illustrated in Yuan et al., 2021, is a very optimistic case); ii) the IGS metadata may be incomplete and iii) the reanalysis may also have breaks. These limitations should be acknowledged in the paper.

Moreover, regarding the first two points, I think the manual approach is very subjective and also probably too conservative. You mention in the former paper that you detected 21 breaks

from 108 stations over 21 years, i.e. an average of 1 break per station every 108 years. This number is very small compared to other studies, e.g. Ning et al., 2016, and Nguyen et al., 2021, using statistical methods. Overall, Nguyen et al., 2021, detected 1 break per station every 5.8 years (after screening) considering all breaks, among which the validated cases represent 1 break per station every 16 years. Both studies also show some obvious examples of undocumented breaks (namely for HERS) and breaks attributed to the reanalysis. Regarding the last point, you write that no obvious breaks were found in the reanalysis. What are your criteria to detect breaks in the reanalysis?

**Reply:** Thank you very much for the comments. The publications you mentioned are very nice and we will cite them. In the revised manuscript, we used all the six atmospheric reanalyses for the detection. For each GPS station, we first tested the changepoints by referring to its log file and identify them by using the detection tool developed by Wang (2008) based on the comparisons of the monthly mean IWV values of GPS and each reanalysis. A changepoint is accepted if it is reported by the tool in at least three GPS–reanalysis comparisons. Its amplitude is calculated as the average of those reported by the comparisons.

Regarding the undocumented changepoints, we determined them carefully. For each GPS station, we first identified the undocumented changepoints automatically reported by the detection tool based on the IWV monthly mean comparisons of GPS and each reanalysis. If similar changepoints within six months are reported by at least three GPS-reanalysis comparisons, they are considered as the changepoints from the GPS IWV series. Then, they are combined into one at the median month, and its amplitude is calculated as the average of those reported by the comparisons.

The performance of different statistical changepoint detection tools has been assessed by Van Malderen et al., (2020), based on synthetic time series with inserted breakpoints. The authors concluded that the combination of different statistical detection tools, together with the use of the available metadata information on GPS instrumental changes, would be the most valid approach of homogenizing GPS IWV time series. In this study, we did combine the use of a statistical detection tool with the use of metadata. In addition, there are disputes on whether the breaks due to extreme climate events or unknown reason should be identified as changepoints. In the revised manuscript, we accepted these changepoints, but with caution.

As for the reason why fewer changepoints were reported in Yuan et al. (2021) than your paper is because we found that Power-Low noise are more suitable than the commonly-used AR(1) for the daily GPS–ERA5 IWV time series over Europe. The assumption of AR(1) noise underestimated the uncertainty of the break amplitude, and thus identify more breaks. Moreover, in that paper, we excluded the first several years in the time series if there is or quality problem. For instance, the data at station HERS before 2001 September was excluded because its quality had been poor until the antenna was repaired at that time (IGSMAIL-3503). However, three changepoints were identified at station HERS during 1998−2021 in Nguyen et al. (2021). In addition, we did not find report of changepoints at station HERS in Ning et al. (2016).

Regarding potential changepoints in reanalyses, we will remind the readers as suggested. By using all reanalyses in the changepoint detection tool, we hope to minimize the effect of these changepoints on the results here, although it cannot completely rule out that identical changepoints appear in different reanalyses by ingesting the same observational datasets through data assimilation. This limitation will be mentioned. However, we do not want to homogenise the reanalyses in this work. This is because they represent the native quality of the reanalyses that we would like to assess.

**Reference**

Ning, T., Wickert, J., Deng, Z., Heise, S., Dick, G., Vey, S., and Schöne, T.: Homogenized Time Series of the Atmospheric Water Vapor Content Obtained from the GNSS Reprocessed Data, Journal of Climate, 29, 2443–2456, https://doi.org/10.1175/JCLI-D-15-0158.1, 2016.

Van Malderen, R., Pottiaux, E., Klos, A., Bock, O., Bogusz, J., Chimani, B., Elias, M., Gruszczynska, M., Guijarro, J., Kazancı, S. Z., and Ning, T.: Homogenizing Gps Integrated Water Vapour Time Series: Methodology and Benchmarking the Algorithms on Synthetic Datasets, 13, 2020.

Wang, X. L.: Penalized Maximal F Test for Detecting Undocumented Mean Shift without Trend Change, Journal of Atmospheric and Oceanic Technology, 25, 368–384, https://doi.org/10.1175/2007JTECHA982.1, 2008.

Yuan, P., Hunegnaw, A., Alshawaf, F., Awange, J., Klos, A., Teferle, F. N., and Kutterer, H.: Feasibility of ERA5 integrated water vapor trends for climate change analysis in continental Europe: An evaluation with GPS (1994–2019) by considering statistical significance, Remote Sensing of Environment, 260, 112416, https://doi.org/10.1016/j.rse.2021.112416, 2021.

**3.** The analysis of the diurnal cycle is interesting. However, to make a fair intercomparison, the reanalyses should be analysed at the smaller common resolution which is 6-hourly, and not interpolated to a higher resolution (1-hourly). For the two reanalyses which have higher resolution (ERA5 and MERRA-2), you may show both the native and under-sampled (6-hourly) results.

**Reply:** Thank you for your constructive suggestion. We modified the comparisons accordingly. We compared the six reanalyses to GPS IWV at the least common temporal resolution (6-hourly). For ERA5 and MERRA2, we also carried out the comparisons at their respective native temporal resolutions. In addition, we would like to evaluate the potential benefit of ERA5 from its improvement in temporal resolution. Therefore, we compared the 1-hourly ERA5 to the other 3-/6-hourly reanalyses in modelling 1-hourly diurnal cycle and intraday variations of IWV.

**4.** In section 3.2, you may mention that the moist bias of ERAI over Europe was also reported by Parracho et al. 2018.

**Reply:** Added as suggested.

**5.** Please explain how you compute the trends.

**Reply:** Added as suggested.

**6.** In Section 6, you may mention that the trend results are also in line with the findings of Parracho et al. 2018, and Nguyen et al., 2021.

**Reply:** Added as suggested.

**7.** What is MERRA2' in Figure 5?

**Reply:** It was a typo and was removed.

Nguyen KN, Quarello A, Bock O, Lebarbier E. Sensitivity of Change-Point Detection and Trend Estimates to GNSS IWV Time Series Properties. Atmosphere. 2021; 12(9):1102. https://doi.org/10.3390/atmos12091102

Pacione, R., Araszkiewicz, A., Brockmann, E., and Dousa, J.: EPN-Repro2: A reference GNSS tropospheric data set over Europe, Atmos. Meas. Tech., 10, 1689–1705, https://doi.org/10.5194/amt-10-1689-2017, 2017.

---

## Author Comment (AC2)

**Reviewer #1: https://doi.org/10.5194/acp-2021-797-RC1**

This manuscript investigated the multi-temporal-scale variabilities and trends of IWV and assessed six commonly-used atmospheric reanalyses (CFSR, ERA5, ERA-Interim, JRA55, MERR2, and NCEP2) over Europe using IWV time series from 108 GPS stations for more than two decades. I have the following comments:

**Reply:** Thanks a lot for your comments.

**Main comments:**

**1.** The authors have taken into account of vertical IWV adjustment. However, the height system of GPS is different from that of the reanalyses. I'm not sure if the authors have considered the unification and differences of the different height systems?

**Reply:** You are right. Ellipsoidal height is usually used in GPS data processing, whereas geopotential height is used in atmospheric reanalyses. To calculate meteorological variables from reanalyses at the location of a GPS station, geopotential height of the station should be used. In this study, the height system conversion was carried out as follows:

We first converted the reference of the GPS station's altitude from ellipsoid to Mean Sea Level (MSL) by using geoid model Earth Gravitational Model 2008 (EGM2008; Pavlis et al., 2012):

$$H_{or} = H_{el} - N, \tag{1}$$

where the $H_{or}$ and $N$ are the orthometric and geoid heights in metre, respectively.

We then adjusted the altitude of the GPS station (with a latitude of $\varphi$) by considering gravity variations (Dirksen et al., 2014; Wang et al., 2016; World Meteorological Organization, 2018):

$$H_{gp} = \frac{\gamma_s(\varphi)}{9.80665} \cdot \frac{R(\varphi) \cdot H_{or}}{R(\varphi) + H_{or}}, \tag{2}$$

$$\gamma_s(\varphi) = 9.780325 \frac{1 + 1.93185 \times 10^{-3} \cdot \sin(\varphi)^2}{(1 - 6.69435 \times 10^{-3} \cdot \sin(\varphi)^2)^{0.5}}, \tag{3}$$

$$R(\varphi) = \frac{6.378137 \times 10^6}{1.006803 - 6.706 \times 10^{-3} \cdot \sin(\varphi)^2}. \tag{4}$$

**Reference**

Dirksen, R. J., Sommer, M., Immler, F. J., Hurst, D. F., Kivi, R., and Vömel, H.: Reference quality upper-air measurements: GRUAN data processing for the Vaisala RS92 radiosonde, Atmos. Meas. Tech., 7, 4463–4490, https://doi.org/10.5194/amt-7-4463-2014, 2014.

Pavlis, N. K., Holmes, S. A., Kenyon, S. C., and Factor, J. K.: The development and evaluation of the Earth Gravitational Model 2008 (EGM2008), Journal of Geophysical Research: Solid Earth, 117, https://doi.org/10.1029/2011JB008916, 2012.

Wang, X., Zhang, K., Wu, S., Fan, S., and Cheng, Y.: Water vapor-weighted mean temperature and its impact on the determination of precipitable water vapor and its linear trend, Journal of Geophysical Research: Atmospheres, 121, 833–852, https://doi.org/10.1002/2015JD024181, 2016.

World Meteorological Organization: Guide to Instruments and Methods of Observation, Measurement of Meteorological Variables. (WMO-No. 8), 1, 2018.

**2.** In the manuscript, the authors used the difference time series between ERA5 IWV and GPS IWV to visually detect the breaks in GPS IWV, so the potential significant differences may be eliminated since the homogenization, also this may be the reason why the ERA5 outperforms than other reanalyses. Are these breaks based on ERA5 IWV still significant, are there any other reanalyses used for the homogenization process?

**Reply:** We agree with you that using ERA5 alone for the homogenization of GPS IWV is unfair to the evaluation of the other five reanalyses. Therefore, in the revised manuscript, we used all the six atmospheric reanalyses for the detection. The procedure is as follows. For each GPS station, we first tested the changepoints by referring to its log file and identify them by using the detection tool developed by Wang (2008) based on the comparisons of the monthly mean IWV values of GPS and each reanalysis. A changepoint is accepted if it is reported by the tool in at least three GPS–reanalysis comparisons. Its amplitude is calculated as the average of those reported by the comparisons.

Regarding the undocumented changepoints, we determined them carefully. For each GPS station, we first identified the undocumented changepoints automatically reported by the detection tool based on the IWV monthly mean comparisons of GPS and each reanalysis. If similar changepoints within six months are reported by at least three GPS-reanalysis comparisons, they are considered as the changepoints from the GPS IWV series. Then, they are combined into one at the median month, and its amplitude is calculated as the average of those reported by the comparisons. By using all reanalyses in the changepoint detection tool, we hope to minimize the effect of these changepoints on the results here, although it cannot completely rule out that identical changepoints appear in different reanalyses by ingesting the same observational datasets through data assimilation. This limitation has been mentioned.

**Reference**

Wang, X. L.: Penalized Maximal F Test for Detecting Undocumented Mean Shift without Trend Change, Journal of Atmospheric and Oceanic Technology, 25, 368–384, https://doi.org/10.1175/2007JTECHA982.1, 2008.

**3.** The spatial resolution contributes to most of the representativeness differences, such as the ERAI provides the products with higher spatial resolution (i.e. 0.25°) than the product used in this paper (0.75°). The conclusion that ERA5 has the best performance on the representativeness differences is questionable. This needs more clarification or convincing statements.

**Reply:** It is possible to download ERAI data at 0.25° spatial resolution, though its native resolution is 0.75°. However, the 0.25° ERAI is obtained from a bilinear interpolation of its native

spatial resolution of 0.75°. Therefore, using 0.25° instead of 0.75° for the ERAI data will not really bring any benefit. An explanation from ECMWF is quoted as follows in italic type:

*https://confluence.ecmwf.int/display/CKB/Does+downloading+data+at+higher+resolution+improve+the+output*

***Does downloading data at higher resolution improve the output?***

*When you download CAMS data, C3S data and other data from ECMWF, you can obtain the output data on its archived grid or on a Cartesian lat/long grid at a custom resolution.*

*You can specify a higher output resolution than the archived resolution, but the resulting data will not contain any more information than the original, it has merely been interpolated[1] to a higher resolution. This makes the output look smoother, but does not increase the accuracy or the precision of the data. However, if you choose to interpolate to a coarser resolution than the archived resolution you should be aware that the data can be aliased, unless care was taken to avoid this.*

*For **ERA-Interim** atmospheric data the point interval on the native Gaussian grid is about 0.75 degrees. You can specify a custom grid on the data server web interface, or using the ECMWF WebAPI or using the MARS client (if you have access to it). On the web interface the default grid for ERA-Interim is lat/long, with a default resolution of 0.75x0.75 degrees (about 80km), approximating the irregular grid spacing on the native Gaussian grid.*

*For **ERA5** HRES atmospheric data the point interval on the native Gaussian grid is about 0.28 degrees. You can download ERA5 data using Python and specify a custom grid and resolution in your script. You should set the horizontal resolution to slightly lower than 0.28 degrees (about 30km), for example to 0.25 degrees, approximating the irregular grid spacing on the native Gaussian grid.*

*[1] When data is interpolated, all continuous fields (e.g. precipitation, temperature) are interpolated by bilinear interpolation, and discrete fields (e.g. vegetation, precipitation type, soil type) and Wave 2D spectra are interpolated by nearest-neighbour. For more information about our grids and interpolations see in this presentation https://confluence.ecmwf.int/download/attachments/55122669/intro-interpolation-2016.pdf?api=v2*

**4.** Line 202: "The 3- and 6-hourly IWVs are linearly interpolated into 1-hourly time series." Have the authors assessed the accuracy of the interpolated IWV? For IWV which changes in a high frequency, linear interpolation seems to be not a good choice.

**Reply:** This is an interesting question. We evaluated four interpolation approaches provided by MATLAB (https://ww2.mathworks.cn/help/matlab/ref/interp1.html?lang=en), including "linear", "spline", "pchip", and "makima". We took the 1-hourly GPS IWV of the 108 stations in Europe as reference series. We then selected two subsets of 3- and 6-hourly GPS IWV series from the 1-hourly series. After that, we interpolated the 3- and 6-hourly GPS IWV

series into 1-hourly series by using the four approaches. The average Root-Mean-Square (RMS) estimates of the IWV differences between the original 1-hourly GPS IWV and those interpolated from 3- and 6-hourly series are as follows:

**Table.** Average RMS of differences between the original 1-hourly GPS IWV (kg m$^{-2}$) and those interpolated from 3- and 6-hourly time series

|          | linear | spline | pchip | makima |
|----------|--------|--------|-------|--------|
| 3-hourly | 0.32   | 0.27   | 0.29  | 0.28   |
| 6-hourly | 0.72   | 0.70   | 0.70  | 0.70   |

The Table shows that "spline" has the lowest average RMS of IWV differences, and we therefore selected "spline" for the temporal interpolation of IWV instead of "linear".

**5.** Line 157: There seems to be a missing full stop between "reanalyses" and "Compared". Please check it.

**Reply:** Corrected.

---

## Author Comment (AC3)

**Reviewer #2: https://doi.org/10.5194/acp-2021-797-RC2**

**General Comments**

The work presented in the manuscript gives an overall summary of applications of ground-based GPS observations in Europe of estimated time series of integrated water vapour (IWV), which to my knowledge is unique. It is broad in the sense that it deals with temporal scales from sub-daily to decades, while many previously published results often focus on one particular "signal", e.g. diurnal, annual, trends. As far as I can tell there are no new results in the manuscript, i.e. results that are different from what is already published. Three times it is stated that the results are "in line" with previously published results (lines 222, 323, and 399). Of course, it is also an important part of research to verify earlier findings, but if possible, I would appreciate if there was more emphasis on noted differences compared to earlier results. I am afraid I cannot help with the details. It is an impressive reference list and for me it is impossible to get a reasonably complete overall knowledge during the time allowed for the review.

**Reply:** Thank you very much for your affirmation.

Ground-based GPS is a unique technique to evaluate the quality of IWV from atmospheric reanalyses. The evaluation can provide information on how to improve their performances in retrieving IWV. However, most previous studies in Europe only evaluated the IWV from ERA-Interim produced by ECMWF, which has been superseded by ERA5 since 2019 August. The time length of those studies are also relatively short (<20 years).

To our knowledge, this is the first study which used 25 years of 1-hourly GPS IWV in Europe to evaluate the performances of the newly released ERA5 in modelling multiple temporal scale variations of IWV from intraday to decades. In addition to the ERA5 and ERA-Interim produced by ECMWF, this study also evaluated the IWV from four commonly used products developed by USA and Japan, which have rarely been evaluated in Europe. An advantage of this comprehensive evaluation is that it is capable to avoid impacts due to differences in reference GPS IWV data and evaluation methods, so that the comparisons on the performances of reanalyses are fair.

We believe the results, especially the evaluation of 1-hourly IWV, are very new and interesting to the community. This is because one of the most important advantages of ERA5 is its much higher temporal resolution compared to the other products (1-hourly v.s. 6-hourly), but its possible improvement has not been evaluated in Europe. GPS IWV is a unique data source to evaluate the 1-hourly ERA5 IWV, as it is famous for its high temporal resolution and high accuracy. Europe, especially its northern part, is characterised with unstable weather condition. Hence, it is a good study region to evaluate the performances of the reanalyses in modelling the high frequency variations of IWV. In this revised manuscript, we carried out more investigations on the intraday variations and diurnal cycles of IWV as suggested, such as the differences in inland and coast.

Moreover, Europe is known as the continent with the most significant warming speed. Evaluations of the long-term IWV trends from the atmospheric reanalyses with the 25 years of GPS IWV are also conducive to a better understanding of climate change.

**Specific comments**

**1.** L108: I do not understand the meaning of "integration rate of 95 %"? Can you explain what is being integrated?

**Reply:** We defined the integration rate of the daily IWV series at a GPS station as follows:

$$rate = \frac{N}{MJD_{last} - MJD_{first} + 1} \times 100\% \qquad (1)$$

where $N$ is the number of daily IWV estimates of the GPS station. $MJD_{first}$ and $MJD_{last}$ are the Modified Julian Dates of the first and last daily IWV estimates, respectively. We added this information to the revised manuscript.

**2.** L112: You report that the observations were weighted based on the elevation angle. Is it not important how the weighting was done (a weighting function including sine and cosine terms)?

**Reply:** Yes, we should provide more details on the GPS data processing. An elevation ($e$) dependent weighting function of $\sin e$ is adopted in the processing in addition to a cut-off elevation angle of 7°.

**3.** L192: It is mentioned that homogenisation was done as described by Yuan et al. (2021). I think such a process is critical and it deserves some more detail in your paper instead of having to go through the reference. For example, do you allow breaks to be inserted in the GPS IWV time series at a specific time epoch even if there has been no change noted in the log file for the hardware or the environment at the site?

**Reply:** As Reviewer #1 and Olivier Bock in their comment were questioning the homogenisation procedure used in the previous version of the manuscript, we decided to extend more on the homogenization and used a statistical changepoint detection tool (Wang, 2008) in combination with the available log file information. Moreover, the detection tool has been applied on the IWV monthly mean differences between GPS and all six reanalyses.

The procedure is as follows. For each GPS station, we first tested the changepoints by referring to its log file and identify them by using the detection tool developed by Wang (2008) based on the comparisons of the monthly mean IWV values of GPS and each reanalysis. A changepoint is accepted if it is reported by the tool in at least three GPS–reanalysis comparisons. Its amplitude is calculated as the average of those reported by the comparisons.

Regarding the undocumented changepoints, we determined them carefully. For each GPS station, we first identified the undocumented changepoints automatically reported by the detection tool based on the IWV monthly mean comparisons of GPS and each reanalysis. If similar changepoints within six months are reported by at least three GPS-reanalysis comparisons, they are considered as the changepoints from the GPS IWV series. Then, they are combined into one at the median month, and its amplitude is calculated as the average of those reported by the comparisons. By using all reanalyses in the changepoint detection tool, we hope to minimize the effect of these changepoints on the results here, although it cannot completely rule out that identical changepoints appear in different reanalyses by ingesting the same observational datasets through data assimilation. This limitation has been mentioned.

**Reference**

Wang, X. L.: Penalized Maximal F Test for Detecting Undocumented Mean Shift without Trend Change, Journal of Atmospheric and Oceanic Technology, 25, 368–384, https://doi.org/10.1175/2007JTECHA982.1, 2008.

**4.** L262: My interpretation is that you determine the amplitudes of the diurnal signal as the peak-to-peak value regardless of when the peaks occur. This makes me wonder if the results will be different if instead the phase and amplitud of the sine wave with a 24 h period is estimated, e.g, through the method of least squares. (In some studies also a semidiurnal term, a period of 12 h, is estimated.) It will be of interest if you comment on this, at least for a couple of sites in different climate zones?

**Reply:** Yes, there are harmonic analysis on the diurnal cycle of IWV (e.g., Steinke et al., 2019). We modelled the diurnal cycle with sine wave and compared to the peak-to-peak estimates as suggested. We also investigated the characteristics of several different climate zones. We are sure that this investigation will provide new results to the community.

**Reference**

Steinke, S., Wahl, S., and Crewell, S.: Benefit of high resolution COSMO reanalysis: The diurnal cycle of column-integrated water vapor over Germany, Meteorologische Zeitschrift, 165–177, https://doi.org/10.1127/metz/2019/0936, 2019.

**5.** L268: You find a correlation between the diurnal amplitude and the station height. Since station height (I guess) correlate with the site's distance to the ocean, another approach would be to correlate the amplitude with this distance. It is well known that the ocean (as long as there is no ice) acts like a low pass filter on daily variations in temperature and humidity.

**Reply:** This is a very interesting idea. We compared the impact of the difference between inland and coast on the diurnal amplitude of IWV as suggested.

**6.** L315: This whole section seems questionable if it is worth to be published? Do the GPS IWV data yield any new findings? Given the very high correlation between IWV from GPS and from the reanalyses, it seems as all the reported patterns, and their time dependences, will be seen by using reanalyses data only?

**Reply:** We agree with you that this part is out of the scope of this work, and thus it was removed. We will address the issues related to interannual variations of IWV in the future. The findings here are interesting, as very a few studies have investigated the teleconnections between the interannual variations of IWV in Europe and various climate indices. Furthermore, although the interannual patterns of IWV can be seen in the reanalyses, their performances can be validated by using GPS IWV measurements.

**Technical Corrections**

**7.** Line (L)1+: You use the American spelling of vapour, although ACP is a European journal?

**Reply:** Thank you for the suggestion. We used "vapour" and the style of English (UK) in the revised manuscript.

**8.** L97: ... IWV -using ...   ?

**Reply:** Replaced with "by using".

**9.** L17: 2%-18%  -->  2 %–18 % (similar changes to be carried out many times in the manuscript)

**Reply:** Replaced hyphen with en dash.

**10.**   L154: IWVs  --> The IWV values ?

**Reply:** Replaced as suggested.

**11.**   L157: reanalyses Compared  -->  reanalyses. Compared

**Reply:** Modified as suggested.

**12.**   L203: IWVs are -->  IWV for all sites and days are ?

**Reply:** Modified as "The daily mean IWV time series of each station is further aggregated into monthly mean IWV series…".

**13.**   L398: (29.5°E, 40.8°N),  -->  (29.5 °E, 40.8 °N),  (see also L447-448)

**Reply:** Modified as suggested.

**14.**   L444: 0-0,4  -->  0.0 – 0.4 ?

**Reply:** Replaced hyphen with en dash.

**15.**   L446: 0,4-1  -->  0.4 – 1.0 ?

**Reply:** Replaced hyphen with en dash.

**16.**   L480+: doi links are missing for almost all references and the established standard acronyms for journals are not used.

**Reply:** Added the doi links and used standard acronyms for journals.

**17.**   Figure 2: The yellow colour is not ideal. I suggest to use cyan or magenta instead. You may also consider to use darker colours in Figures 5 and 8. Different colours in these figures are not really needed for clarity, although it may look nicer compared to have it all in black.

**Reply:** We used cyan to replace the yellow colour as suggested. We also used darker colours in Figure 5 and 8.

---

## Author Response (AR1)

**Community #1: https://doi.org/10.5194/acp-2021-797-CC1**

Dear Peng Yuan and co-authors,

Thank you for releasing this interesting study. I am happy to see that you used the representativeness statistic that we proposed in a previous publication and that you confirm and extend our results to other reanalyses.

Below I submit a few questions and comments about your manuscript. Thank you in advance for your answers.

Best regards,

Olivier BOCK

**Reply:** Thanks a lot for your comments. **Our replies are shown in blue. The related texts copied from the revised manuscript are shown in green.**

**1.** Please comment on the choice and on the quality of the used GPS data set (NGL), as other data sets exist for Europe (e.g. the EPN repro2, Pacione et al., 2017).

**Reply:** Pacione et al. (2017) developed a GNSS tropospheric dataset of about 18 years (1996–2014) in the framework of EPN-Repro2. The authors also compared the results obtained with different GNSS data processing schemes. In this work, we selected the ZTD product developed by NGL as it has a much longer time series from 1994 Jan. to 2018 Dec. (25 years). As IWV is the meteorological parameter that we are interested in, we directly evaluated GPS IWV instead of ZTD. Our comparisons show that the IWV estimates from NGL's ZTD and various reanalyses are in good agreement, indicating that the ZTD product has good quality.

**2.** Please provide more details on the homogenization method and results (e.g. the number and magnitude of detected breaks) and comment on their uncertainty. Explain also how the offsets in the GPS series are corrected, knowing that the breaks are detected in the GPS – reanalysis series and not in the GPS series directly.

Regarding the homogenization method, I checked your earlier paper (Yuan et al., 2021), and was wondering why you used a manual segmentation method when many statistical methods exist, which have been assessed by Van Malderen et al., 2020. Can you comment on that choice?

I also understand that in your segmentation method, you select only breaks which are confirmed by known equipment changes from the IGS log files. As you may have experienced: i) not all breaks are easy to detect (the example illustrated in Yuan et al., 2021, is a very optimistic case); ii) the IGS metadata may be incomplete and iii) the reanalysis may also have breaks. These limitations should be acknowledged in the paper.

Moreover, regarding the first two points, I think the manual approach is very subjective and also probably too conservative. You mention in the former paper that you detected 21 breaks from 108 stations over 21 years, i.e. an average of 1 break per station every 108 years. This number is very small compared to other studies, e.g. Ning et al., 2016, and Nguyen et al., 2021,

using statistical methods. Overall, Nguyen et al., 2021, detected 1 break per station every 5.8 years (after screening) considering all breaks, among which the validated cases represent 1 break per station every 16 years. Both studies also show some obvious examples of undocumented breaks (namely for HERS) and breaks attributed to the reanalysis. Regarding the last point, you write that no obvious breaks were found in the reanalysis. What are your criteria to detect breaks in the reanalysis?

**Reply:** Thank you very much for the comments. The publications you mentioned are very instructive and we cited them. In the revised manuscript, we described the homogenisation approach briefly in Section 2.5 Homogenisation, and provided 
[revised manuscript text omitted]

**3.** The analysis of the diurnal cycle is interesting. However, to make a fair intercomparison, the reanalyses should be analysed at the smaller common resolution which is 6-hourly, and not interpolated to a higher resolution (1-hourly). For the two reanalyses which have higher resolution (ERA5 and MERRA-2), you may show both the native and under-sampled (6-hourly) results.

**Reply:** Thank you for your constructive suggestion. We modified the comparisons accordingly. We first evaluated all the reanalyses with respect to GPS at 1-hour temporal resolution. For the reanalyses with coarser resolutions (3-hour and 6-hour), we interpolated their time series to 1-hour by using cubic spline, which was found to be slightly superior to linear interpolation (see our reply to *Reviewer#1 Comment #4*).

For a fairer intercomparison, we also evaluated ERA5 and the other reanalyses at their respective native resolutions. Accordingly, we extracted the ERA5 IWV every 3 and 6 hours. Statistics of the evaluation results are listed in Table 2.

**Table 2.** Statistics of the consistencies in diurnal IWV anomalies from reanalyses compared to GPS.

| | Temporal resolution | CFSR | ERA5 | ERAI | JRA55 | MERRA2 | NCEP2 |
|---|---|---|---|---|---|---|---|
| $\gamma$ | 1 h | 0.98±0.04 | 0.98±0.03 | 0.91±0.04 | 0.86±0.04 | 1.02±0.04 | 0.75±0.05 |
| | 3 h | − | 0.97±0.03 | − | − | 1.02±0.04 | − |
| | 6 h | 0.99±0.03 | 0.96±0.03 | 0.92±0.04 | 0.87±0.04 | − | 0.76±0.05 |
| $r$ | 1 h | 0.85±0.05 | 0.89±0.05 | 0.85±0.06 | 0.83±0.06 | 0.86±0.06 | 0.69±0.08 |
| | 3 h | − | 0.89±0.05 | − | − | 0.86±0.06 | − |
| | 6 h | 0.87±0.05 | 0.90±0.06 | 0.86±0.07 | 0.84±0.07 | − | 0.69±0.08 |
| KGE | 1 h | 0.85±0.05 | 0.88±0.06 | 0.82±0.07 | 0.78±0.07 | 0.85±0.06 | 0.60±0.09 |
| | 3 h | − | 0.89±0.06 | − | − | 0.86±0.06 | − |
| | 6 h | 0.86±0.06 | 0.89±0.06 | 0.83±0.07 | 0.79±0.07 | − | 0.60±0.08 |
| $RMS_\Delta$ (kg m$^{-2}$) | 1 h | 1.14±0.19 | 0.97±0.21 | 1.14±0.22 | 1.20±0.21 | 1.14±0.22 | 1.57±0.24 |
| | 3 h | − | 0.97±0.21 | − | − | 1.15±0.22 | − |
| | 6 h | 1.09±0.21 | 0.94±0.21 | 1.09±0.24 | 1.16±0.22 | − | 1.59±0.24 |

**4.** In section 3.2, you may mention that the moist bias of ERAI over Europe was also reported by Parracho et al. 2018.

**Reply:** Added as suggested.

**5.** Please explain how you compute the trends.

**Reply:** We redrafted Section 6 Assessments of linear trends and provided more details on the calculation of IWV trends.

**In Section 6 Assessments of linear trends**

We carried out a homogenisation of the GPS IWV time series by using the RHtestsV4 software (Wang and Feng, 2013) before the analysis. The software is dedicated to the homogenisation of climatic time series. In addition, we adopted a homogenisation strategy which allows for changepoints with and without support from metadata. We took all the six reanalyses as references and attempted to avoid the impacts of possible changepoints in specific reanalyses. However, we did not homogenise the reanalyses IWV time series because they represent the native quality of the reanalyses that we would like to assess. The homogenisation approach is detailed in Appendix A.

We then estimated the linear IWV trends from all the six reanalyses and the homogenised GPS IWV time series after the removal of annual cycle. In order to obtain realistic uncertainties of the trend estimates, we analysed the time series by using the Hector software version 1.7.2 (Bos et al.,2019). We tested four commonly used noise models, namely White Noise (WN), first-order AutoRegressive AR(1), AutoRegressive Moving Average ARMA(1,1), and Power-Law noise (PL). We then selected the optimal model of each time series by using Bayesian information criterion (BIC; Schwarz, 1978). Readers are referred to Yuan et al. (2021) for more details. The IWV trend estimates, associated uncertainties, and specific optimal noise models are listed in Table S3.

**6.** In Section 6, you may mention that the trend results are also in line with the findings of Parracho et al. 2018, and Nguyen et al., 2021.

**Reply:** Added as suggested.

**In Section 6 Assessments of linear trends**

the geographical patterns of the IWV trends are consistent to previous studies (Parracho et al., 2018; Nguyen et al., 2021; Yuan et al., 2021).

**7.** What is MERRA2' in Figure 5?
**Reply:** It was a typo and was removed.

**Reviewer #1: https://doi.org/10.5194/acp-2021-797-RC1**

This manuscript investigated the multi-temporal-scale variabilities and trends of IWV and assessed six commonly-used atmospheric reanalyses (CFSR, ERA5, ERA-Interim, JRA55, MERR2, and NCEP2) over Europe using IWV time series from 108 GPS stations for more than two decades. I have the following comments:

**Reply:** Thanks a lot for your comments. **Our replies are shown in blue**. **The related texts copied from the revised manuscript are shown in green.**

**Main comments:**

**1.** The authors have taken into account of vertical IWV adjustment. However, the height system of GPS is different from that of the reanalyses. I'm not sure if the authors have considered the unification and differences of the different height systems?

**Reply:** You are right. Ellipsoidal height is usually used in GPS data processing, whereas geopotential height is used in atmospheric reanalyses. To calculate meteorological variables from reanalyses at the location of a GPS station, the geopotential height of the station should be used. We added the following texts.

**In section 2.3 IWV retrievals**

It is noteworthy that geopotential height system is employed in the reanalyses. Accordingly, the geopotential heights ($H_{gp}$) of the GPS stations rather than their ellipsoidal ($H_{el}$) or orthometric heights ($H_{or}$) are used in the above calculations. The conversion of height systems is carried out as follows (Dirksen et al., 2014; Wang et al., 2016; World Meteorological Organization, 2018):

$$H_{or} = H_{el} - N,$$

$$H_{gp} = \frac{\gamma_s(\varphi_s)}{9.80665} \cdot \frac{R(\varphi_s) \cdot H_{or}}{R(\varphi_s) + H_{or}}, \tag{5}$$

$$\gamma_s(\varphi_s) = 9.780325 \frac{1 + 1.93185 \times 10^{-3} \cdot \sin^2(\varphi_s)}{(1 - 6.69435 \times 10^{-3} \cdot \sin^2(\varphi_s))^{0.5}}, \tag{6}$$

$$R(\varphi_s) = \frac{6.378137 \times 10^6}{1.006803 - 6.706 \times 10^{-3} \cdot \sin(\varphi_s)^2}, \tag{7}$$

where the $N$ is the geoid heights in meter from the Earth Gravitational Model 2008 (EGM2008; Pavlis et al., 2012).

**Reference**

Dirksen, R. J., Sommer, M., Immler, F. J., Hurst, D. F., Kivi, R., and Vömel, H.: Reference quality upper-air measurements: GRUAN data processing for the Vaisala RS92 radiosonde, Atmos. Meas. Tech., 7, 4463–4490, https://doi.org/10.5194/amt-7-4463-2014, 2014.

Pavlis, N. K., Holmes, S. A., Kenyon, S. C., and Factor, J. K.: The development and evaluation of the Earth Gravitational Model 2008 (EGM2008), J. Geophys. Res.: Solid Earth, 117, https://doi.org/10.1029/2011JB008916, 2012.

Wang, X., Zhang, K., Wu, S., Fan, S., and Cheng, Y.: Water vapor-weighted mean temperature and its impact on the determination of precipitable water vapor and its linear trend, J. Geophys. Res.: Atmos., 121, 833–852, https://doi.org/10.1002/2015JD024181, 2016.

World Meteorological Organization: Guide to Instruments and Methods of Observation, Measurement of Meteorological Variables. (WMO-No. 8), 1, 2018.

**2.** In the manuscript, the authors used the difference time series between ERA5 IWV and GPS IWV to visually detect the breaks in GPS IWV, so the potential significant differences may be eliminated since the homogenization, also this may be the reason why the ERA5 outperforms than other reanalyses. Are these breaks based on ERA5 IWV still significant, are there any other reanalyses used for the homogenization process?

**Reply:** We agree with you that using ERA5 alone for the homogenization of GPS IWV is unfair to the evaluation of the other five reanalyses. Therefore, in the revised manuscript, we used all the six atmospheric reanalyses for the detection. Moreover, we described the homogenisation approach briefly in Section 2.5 Homogenisation and provided step-by-step details and examples at two stations (HERS and ERLA) in Appendix A: Homogenisation of GNSS IWV time series. You can also find the copied texts in our *reply to Community #1 Comment #2*.

**3.** The spatial resolution contributes to most of the representativeness differences, such as the ERAI provides the products with higher spatial resolution (i.e. 0.25°) than the product used in this paper (0.75°). The conclusion that ERA5 has the best performance on the representativeness differences is questionable. This needs more clarification or convincing statements.

**Reply:** It is possible to download ERAI data at 0.25° spatial resolution, though its native resolution is 0.75°. However, the 0.25° ERAI is obtained from a bilinear interpolation of its native spatial resolution of 0.75°. Therefore, using 0.25° instead of 0.75° for the ERAI data will not really bring any benefit. An explanation from ECMWF is quoted as follows in *italic type*:

https://confluence.ecmwf.int/display/CKB/Does+downloading+data+at+higher+resolution+improve+the+output

*Does downloading data at higher resolution improve the output?*

*When you download CAMS data, C3S data and other data from ECMWF, you can obtain the output data on its archived grid or on a Cartesian lat/long grid at a custom resolution.*

*You can specify a higher output resolution than the archived resolution, but the resulting data will not contain any more information than the original, it has merely been interpolated[1] to a higher resolution. This makes the output look smoother, but does not increase the accuracy or the precision of the data. However, if you choose to interpolate to a coarser resolution than the archived resolution you should be aware that the data can be aliased, unless care was taken to avoid this.*

*For **ERA-Interim** atmospheric data the point interval on the native Gaussian grid is about 0.75 degrees. You can specify a custom grid on the data server web interface, or using the ECMWF WebAPI or using the MARS client (if you have access to it).  On the web interface*

*the default grid for ERA-Interim is lat/long, with a default resolution of 0.75x0.75 degrees (about 80km), approximating the irregular grid spacing on the native Gaussian grid.*

*For **ERA5** HRES atmospheric data the point interval on the native Gaussian grid is about 0.28 degrees. You can download ERA5 data using Python and specify a custom grid and resolution in your script. You should set the horizontal resolution to slightly lower than 0.28 degrees (about 30km), for example to 0.25 degrees, approximating the irregular grid spacing on the native Gaussian grid.*

*[1] When data is interpolated, all continuous fields (e.g. precipitation, temperature) are interpolated by bilinear interpolation, and discrete fields (e.g. vegetation, precipitation type, soil type) and Wave 2D spectra are interpolated by nearest-neighbour. For more information about our grids and interpolations see in this presentation https://confluence.ecmwf.int/download/attachments/55122669/intro-interpolation-2016.pdf?api=v2*

**4.** Line 202: "The 3- and 6-hourly IWVs are linearly interpolated into 1-hourly time series." Have the authors assessed the accuracy of the interpolated IWV? For IWV which changes in a high frequency, linear interpolation seems to be not a good choice.

**Reply:** Thank you for the constructive suggestion. We evaluated four interpolation approaches provided by MATLAB (https://ww2.mathworks.cn/help/matlab/ref/interp1.html?lang=en), including "linear", "spline", "pchip", and "makima". We took the 1-hourly GPS IWV of the 108 stations in Europe as reference series. We then selected two subsets of 3- and 6-hourly GPS IWV series from the 1-hourly series. After that, we interpolated the 3- and 6-hourly GPS IWV series into 1-hourly series by using the four approaches. The average Root-Mean-Square (RMS) estimates of the IWV differences between the original 1-hourly GPS IWV and those interpolated from 3- and 6-hourly series are as follows:

**Table.** Average RMS of differences between the original 1-hourly GPS IWV (kg m$^{-2}$) and those interpolated from 3- and 6-hourly time series

|          | linear | spline | pchip | makima |
|----------|--------|--------|-------|--------|
| 3-hourly | 0.32   | 0.27   | 0.29  | 0.28   |
| 6-hourly | 0.72   | 0.70   | 0.70  | 0.70   |

The Table shows that "spline" has the lowest average RMS of IWV differences, and we therefore selected "spline" for the temporal interpolation of IWV instead of "linear".

We therefore added the following words to Section 4.3 Diurnal anomalies:

For the reanalyses with coarser resolutions (3-hour and 6-hour), we interpolated their time series to 1-hour by using cubic spline, which is slightly superior to linear interpolation.

**5.** Line 157: There seems to be a missing full stop between "reanalyses" and "Compared". Please check it.

**Reply:** Corrected.

**Reviewer #2: https://doi.org/10.5194/acp-2021-797-RC2**

**General Comments**

The work presented in the manuscript gives an overall summary of applications of ground-based GPS observations in Europe of estimated time series of integrated water vapour (IWV), which to my knowledge is unique. It is broad in the sense that it deals with temporal scales from sub-daily to decades, while many previously published results often focus on one particular "signal", e.g. diurnal, annual, trends. As far as I can tell there are no new results in the manuscript, i.e. results that are different from what is already published. Three times it is stated that the results are "in line" with previously published results (lines 222, 323, and 399). Of course, it is also an important part of research to verify earlier findings, but if possible, I would appreciate if there was more emphasis on noted differences compared to earlier results. I am afraid I cannot help with the details. It is an impressive reference list and for me it is impossible to get a reasonably complete overall knowledge during the time allowed for the review.

**Reply:** Thank you very much for your affirmation. **Our replies are shown in blue. The related texts copied from the revised manuscript are shown in green.**

Ground-based GPS is a unique technique to evaluate the quality of IWV from atmospheric reanalyses. The evaluation can provide information on how to improve their performances in retrieving IWV. However, most previous studies in Europe only evaluated the IWV from ERA-Interim produced by ECMWF, which has been superseded by ERA5 since 2019 August. The time lengths of those studies are also relatively short (<20 years).

To our knowledge, this is **the first study** which used 25 years of 1-hourly GPS IWV in Europe to evaluate the performances of the newly released ERA5 in modelling multiple temporal scale variations of IWV from intraday to decades. In addition to the ERA5 and ERA-Interim produced by ECMWF, this study also evaluated the IWV from four commonly used products developed by USA and Japan, which have rarely been evaluated in Europe. An advantage of this comprehensive evaluation is that it is capable to avoid impacts due to differences in reference GPS IWV data and evaluation methods, so that the evaluations on the performances of the various reanalyses are more comparable.

There is an interesting finding which **has never been reported in previous studies** as far as we know. That is, the mismatches in the diurnal cycle of ERA5 IWV from 09 to 10 UTC and from 21 to 22 UTC. The artificial shifts are most likely due to the edge effect in each ERA5 assimilation cycle. We carried out an elaborate analysis on the spatiotemporal characterisations of the artificial shifts in ERA5 IWV diurnal cycles. Results show that they are -0.08 and 0.19 kg m$^{-2}$ at the two epochs and cannot be ignored. We added a specific section for this finding, see Section 4.2 Mismatches in diurnal ERA5 IWV cycle.

We also modified Section 4.1 Diurnal GPS IWV cycle as suggested by your comment #4. We modelled the diurnal IWV cycle by using diurnal and semidiurnal harmonics. Moreover, we analysed the characteristics of the diurnal IWV cycles by considering each station's altitude, and

distance-to-sea, and climate zones. We also discussed different mechanisms, such as solar heating, land-sea breeze, and orographic circulation. This comprehensive analysis **is rare in Europe** as far as we know.

We compared the intraday IWV variations quantified as diurnal anomalies and confirmed that ERA5 is superior to the other reanalyse at all the 1-, 3-, and 6-hour temporal resolutions. We believe the results are very new and interesting to the community. This is because one of the most important advantages of ERA5 is its much higher temporal resolution compared to the other products (1-hourly versus. 3-, 6-hourly), but its possible improvement **has not been evaluated in Europe**.

Europe is known as the continent with the most significant warming speed. Evaluations of the long-term IWV trends from the atmospheric reanalyses with the 25 years of GPS IWV are also conducive to a better understanding of climate change. In the revised manuscript, we modified the homogenisation of GPS IWV time series and carried out fairer intercomparisons and evaluations of the long-term IWV trends from six reanalyses in Europe. We concluded that ERA5 is also superior to the others in modelling the linear IWV trends. However, due to significant discrepancies with respect to GPS, CFSR and NCEP2 are not recommend for the analysis of IWV trends over Southern Europe and the whole Europe, respectively. To our knowledge, these intercomparisons and evaluations are **reported for the first time**.

**Specific comments**

**1.** L108: I do not understand the meaning of "integration rate of 95 %"? Can you explain what is being integrated?

**Reply:** We defined the integration rate of the daily IWV series at a GPS station as follows:

$$rate = \frac{N}{MJD_{last} - MJD_{first} + 1} \times 100\% \tag{1}$$

where $N$ is the number of daily IWV estimates of the GPS station. $MJD_{first}$ and $MJD_{last}$ are the Modified Julian Dates of the first and last daily IWV estimates, respectively. We added the following sentence to Section 2.1 GPS data:

The integration rate is the ratio between the number of available daily IWV data points and the theoretical number of all possible observations in the time range.

**2.** L112: You report that the observations were weighted based on the elevation angle. Is it not important how the weighting was done (a weighting function including sine and cosine terms)?

**Reply:** We added the following sentence to Section 2.1 GPS data:

The observations were weighted based on elevation ($e$) dependent function of $\sin e$. The cutoff elevation angle was set to $7°$.

**3.** L192: It is mentioned that homogenisation was done as described by Yuan et al. (2021). I think such a process is critical and it deserves some more detail in your paper instead of having to go through the reference. For example, do you allow breaks to be inserted in the GPS IWV time series at a specific time epoch even if there has been no change noted in the log file for the hardware or the environment at the site?

**Reply:** We modified the homogenisation approach in the revised manuscript by using a statistical changepoint detection tool, the RHtestsV4 software (Wang and Feng, 2013). The software is based on a penalized maximal *t* test with the consideration of linear trend, annual cycle, and AR(1) noise in the time series (Wang et al., 2007; Wang 2008).

In addition to the documented changepoints in metadata (GPS station log files and IGSMAIL), we also allowed for undocumented changepoints in the GPS IWV time series without support from the metadata. We described the homogenisation approach briefly in Section 2.5 Homogenisation and provided step-by-step details and examples at two stations (HERS and ERLA) in Appendix A: Homogenisation of GNSS IWV time series. You can also find the copied texts in our *reply to Community #1 Comment #2*.

**Reply:** We modified the colour scheme of Fig. 2. We also used darker colours in Figure 5.

[Figure]

**Figure 2.** Schematic plot of the vertical and horizontal interpolation of the reanalysis pressure level products. The IWV at each auxiliary point ($A_i$; orange dots) is calculated with vertical interpolation or extrapolation of the adjacent reanalysis nodes ($N_i^{(j)}$ and $N_i^{(j+1)}$; green dots). The IWV at the GPS station (red dot) is then estimated with horizontal interpolation of the auxiliary points.

[Figure]

**Figure 5.** Box-whisker plots of the KGE parameters for the daily time series (a−d) and monthly annual cycle of IWV (e−h) from the reanalyses compared to GPS for the 108 stations.